# OrthoID: profiling dynamic proteomes through time and space using mutually orthogonal chemical tools

Ara Lee [1,2,12], Gihyun Sung [1,2,12], Sanghee Shin[3,4], Song-Yi Lee [5], Jaehwan Sim[1,6], Truong Thi My Nhung[7], Tran Diem Nghi[7], Sang Ki Park [7], Ponnusamy Pon Sathieshkumar[1], Imkyeung Kang[8,9], Ji Young Mun[8], Jong-Seo Kim [3,4] ✉, Hyun-Woo Rhee [5] ✉, Kyeng Min Park [10] ✉ & Kimoon Kim [1,2,6,11] ✉

Identifying proteins at organelle contact sites, such as mitochondria-associated endoplasmic reticulum membranes (MAM), is essential for understanding vital cellular processes, yet challenging due to their dynamic nature. Here we report "OrthoID", a proteomic method utilizing engineered enzymes, TurboID and APEX2, for the biotinylation (Bt) and adamantylation (Ad) of proteins close to the mitochondria and endoplasmic reticulum (ER), respectively, in conjunction with high-affinity binding pairs, streptavidin-biotin (SA-Bt) and cucurbit[7]uril-adamantane (CB[7]-Ad), for selective orthogonal enrichment of Bt- and Ad-labeled proteins. This approach effectively identifies protein candidates associated with the ER-mitochondria contact, including LRC59, whose roles at the contact site were—to the best of our knowledge—previously unknown, and tracks multiple protein sets undergoing structural and locational changes at MAM during mitophagy. These findings demonstrate that OrthoID could be a powerful proteomics tool for the identification and analysis of spatiotemporal proteins at organelle contact sites and revealing their dynamic behaviors in vital cellular processes.

Cellular organelles communicate to regulate vital cellular processes, including biomolecule exchange, signal transduction, and homeostasis in response to metabolic and environmental cellular conditions. Organelle communication is largely mediated by proteins at membrane contact sites, including mitochondria-associated endoplasmic reticulum membranes (MAM)[1–5]. Thus, identification of

proteins at contact sites is crucial to comprehend the function of and mechanisms underlying organelle communication. However, difficulties in reliable analysis of proteins localized in specific subcellular regions hinder the protein identification process.

Engineered enzyme-mediated proximity labeling (PL), coupled with the naturally occurring high-affinity binding pair of biotin (Bt) and

[1]Center for Self–assembly and Complexity, Institute for Basic Science (IBS), Pohang, Republic of Korea. [2]Division of Advanced Materials Science (AMS), Pohang University of Science and Technology (POSTECH), Pohang, Republic of Korea. [3]Center for RNA Research, Institute for Basic Science (IBS), Seoul, Republic of Korea. [4]School of Biological Sciences, Seoul National University, Seoul, Republic of Korea. [5]Department of Chemistry, Seoul National University, Seoul, Republic of Korea. [6]School of Interdisciplinary Bioscience and Bioengineering, Pohang University of Science and Technology (POSTECH), Pohang, Republic of Korea. [7]Department of Life Sciences, Pohang University of Science and Technology (POSTECH), Pohang, Republic of Korea. [8]Neural Circuit Research Group, Korea Brain Research Institute, Daegu, Republic of Korea. [9]Department of Microbiology, University of Ulsan College of Medicine, Ulsan, Republic of Korea. [10]Department of Biochemistry, Daegu Catholic University School of Medicine, Daegu, Republic of Korea. [11]Department of Chemistry, Pohang University of Science and Technology (POSTECH), Pohang, Republic of Korea. [12]These authors contributed equally: Ara Lee, Gihyun Sung.
✉e-mail: jongseokim@snu.ac.kr; rheehw@snu.ac.kr; kpark@cu.ac.kr; kkim@postech.ac.kr

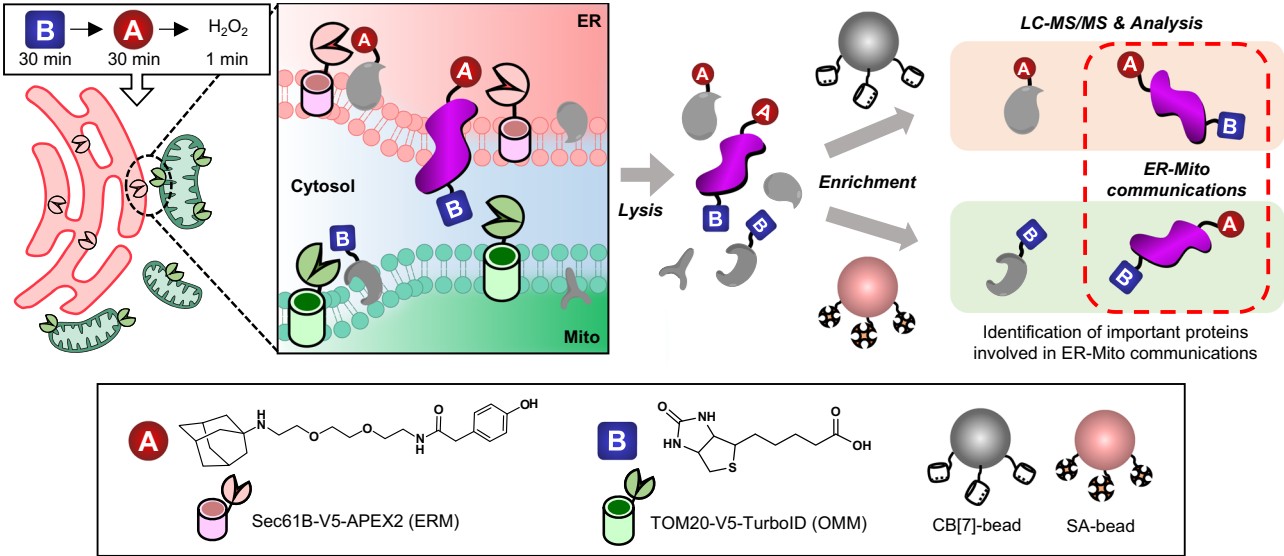

**Fig. 1 | Schematic description of the protein labeling and identification process (OrthoID).** Sec61B-V5-APEX2 and TOM20-V5-TurboID were expressed in ERM and OMM, respectively, for orthogonal labeling of target proteins. Labeled proteins were isolated by CB[7]- and SA-beads in parallel and analyzed by LC-MS/MS.

streptavidin (SA), has significantly contributed to the development of spatial proteomics. Moreover, advances in liquid chromatography-tandem mass spectrometry (LC-MS/MS) enabled the labeled proteins to be specifically analyzed by detecting shifted mass values through Bt conjugation, as is done in the SPOT-BioID method[6]. Notably, tandem mass spectrometry, which detects chemically labeled peptides with high reliability, ensures the selective identification of proteins influenced by the orientation and localization of PL-enzymes in proteomic analysis. This feature significantly diminishes false positives, thereby enhancing the accuracy of spatial proteomics[7]. Promiscuous Bt ligases (BioID and TurboID) and engineered ascorbate peroxidase 2 (APEX2) and their substrates, Bt and phenol-conjugated Bt, have been extensively used for rapid biotinylation of lysine and tyrosine residues, respectively, of their proximal proteins in live cells[8–12]. The proteins at MAM were biotinylated using the full-length or split enzymes expressed on the cytoplasmic face of the outer mitochondrial (OMM) and endoplasmic reticulum (ERM) membranes. These proteins were subsequently identified through LC-MS/MS after enrichment with SA-beads[7,13–15]. The SA-Bt binding pair has provided interesting spatiotemporal proteomic information for PL-based proteomics. However, several issues in the system limit precise proteomic analysis, with unavoidable detections of naturally occurring endogenously biotinylated proteins as false positives[16–18]. Additionally, the observation of undesired substantial biotinylation to cytosolic proteins suggests the need for careful control experiments and complex proteomics data processing[7,15]. Furthermore, practical availability of only one binding pair system for PL and selective protein enriching restricts the employment of multiple enzymes in the same cells and prevents the enhancement of reliability, accuracy, and precision of spatial proteomics.

Cucurbit[7]uril (CB[7]) has been investigated as a synthetic host molecule that forms a high-affinity host-guest complex with its selected guests, including adamantylammonium ($K_a = 10^{13}$ M$^{-1}$, comparable to that of SA-Bt)[19–21]. The unique host-guest interaction has recently been exploited as a non-covalent click system for detection, isolation, and purification of proteins, as well as visualization of cellular processes and animal tumors[22–32]. As APEX-mediated proximity labeling worked well with phenol-conjugated adamantane (Ad) as its substrate, proteins labeled with Ad (Ad-proteins) were selectively visualized with fluorophore-conjugated CB[7], such as cyanine3-conjugated CB[7]

(Cy3-CB[7]), and efficiently isolated using CB[7]-conjugated beads (CB[7]-beads) to analyze cellular proteins[23,32]. CB[7]-Ad works similarly to SA-Bt in terms of high affinity binding. However, their selective binding does not conflict because they have distinct binding mechanisms. This led to our utilization of both binding pairs together along with two engineered enzymes as a mutually orthogonal dual system for spatiotemporal proteomics.

Here, we present OrthoID, a chemical approach for the accurate detection of proteins that are spatiotemporally localized at contact sites of organelles using two different engineered enzymes (TurboID and APEX2) in combination with mutually orthogonal dual high-affinity binding pairs (SA-Bt and CB[7]-Ad) (Fig. 1). Cells expressing APEX2 and TurboID on the lumen-facing ERM and the cytosol-facing OMM, respectively, are exploited for rapid proximity labeling of ER proteins contacting OMM with Ad and Bt. The proteins are subsequently identified through the accurate detection of Ad- and Bt-labeled peptides by LC-MS/MS using the shifted mass values. Ad is detected on a tyrosine residue (SPOT-SupraID)[32] and Bt on a lysine residue (SPOT-BioID)[6] after selective enrichment with CB[7]- and SA-beads, respectively. Comparative analysis of the SPOT-SupraID and SPOT-BioID results leads to the identification of 21 proteins localized between the ER and mitochondria. These proteins include LRC59, TGO1, KPYM, GANAB, and MESD, whose association with ER-mitochondria (ER-mito) communication is previously unknown. Furthermore, this approach enables successful detection of spatiotemporal protein sets that undergo structural and locational changes at MAM upon induction of specific cellular processes, such as mitophagy. The detected protein sets include MTCH2, RACK1, HCD2, and MIC60. Our findings highlight the great potential of this proteomics strategy for the discovery of proteins that are present in specific subcellular compartments at specific times, particularly those involved in organelle contact and communication.

## Results and discussion
### An orthogonal dual fishing system for detection of AdBt-proteins
Prior to the experiments with live cells, the orthogonality of SA-Bt and CB[7]-Ad for the detection of Bt- and Ad-labeled proteins was investigated using model proteins such as Bt-labeled myoglobin (Bt-MYB, 17 kDa), Ad-labeled ovalbumin (Ad-OVA, 43 kDa), and AdBt-

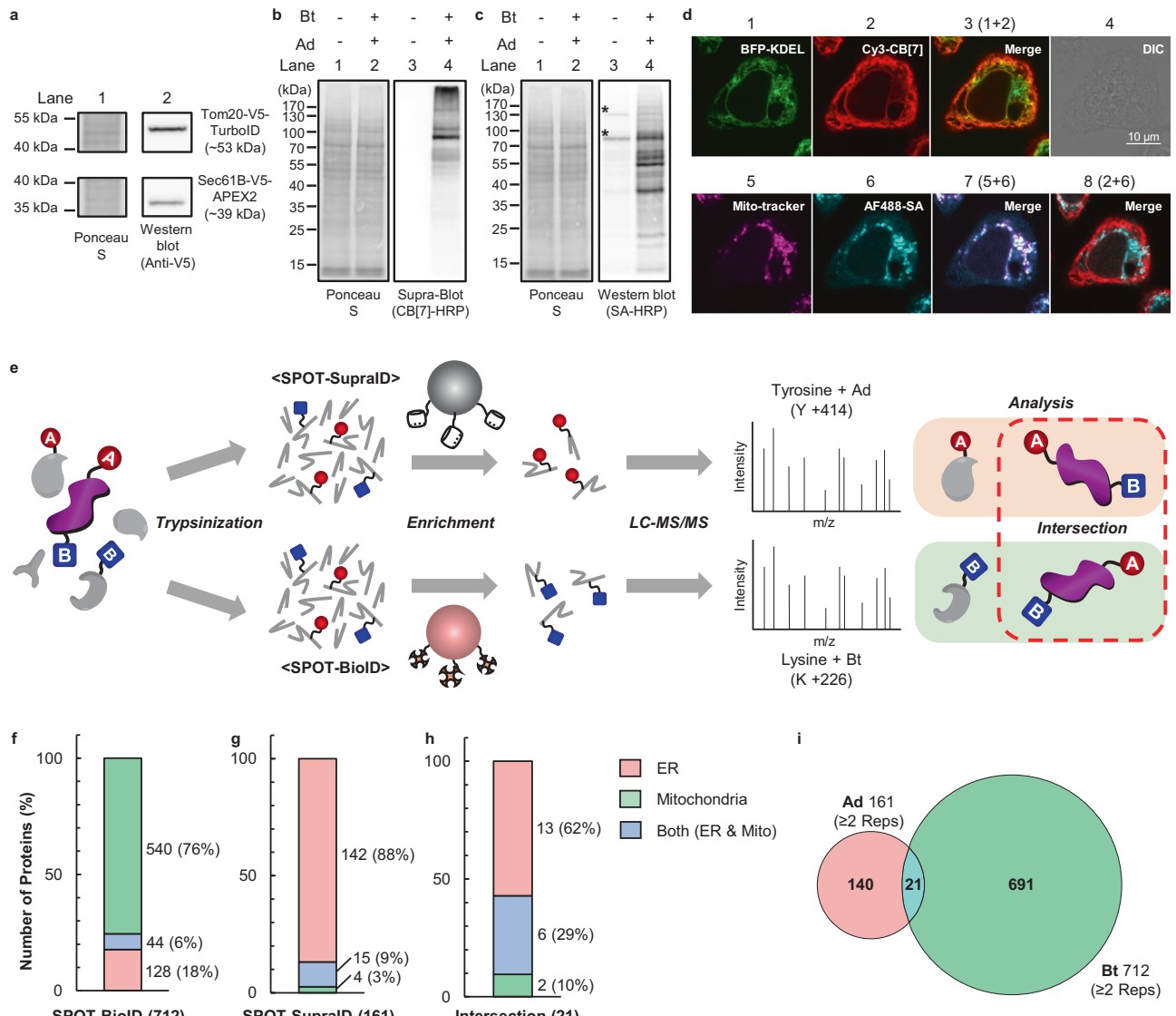

**Fig. 2 | Characterization of dual-enzyme expressed cells and identification of proteins at ER-mito contact sites through OrthoID. a** Western blotting of V5-tagged enzymes. **b**, **c** Supra-blotting **b** for Ad-labeled proteins and western blotting **c** for Bt-labeled proteins (the asterisk (*) represents endogenously biotinylated proteins). **d** CLSM images with BFP-KDEL (green) as an ER-marker, mito-tracker (purple) as a mito-marker, Cy3-CB[7] (red) for Ad-labeled proteins, and AF488-SA (cyan) for Bt-labeled proteins. $r$ (Pearson's coefficient)[72] = 0.66 for image 3, 0.89 for image 7, and 0.51 for image 8. Scale bar: 10 μm. Blots and image data **a**–**d** are representative of at least three independent experiments. **e** A schematic description of OrthoID. **f**–**h** Bar graphs showing the number of total identified proteins from SPOT-BioID, SPOT-SupraID and from the intersection (OrthoID) with ER and mitochondria annotation. Color represents GOCC annotation of proteins; ER (pink), mitochondria (green), and both (blue). **i** A Venn diagram of SPOT-SupraID and SPOT-BioID, 21 proteins found at the intersection (OrthoID). Source data are provided as a Source Data file.

labeled bovine serum albumin (AdBt-BSA, 66 kDa) by LC-MS/MS (See Methods for preparation, Supplementary Fig. 2). Trypsinized peptides from the three model protein mixtures were enriched with SA- and CB[7]-beads and subjected to LC-MS/MS (Supplementary Fig. 3a). Then, detection of peptides with increased mass values on lysine (K + 226.078 Da) through Bt labeling (SPOT-BioID) and tyrosine (Y + 414.252 Da) through Ad labeling (SPOT-SupraID) enabled the identification of labeled proteins using the Proteome Discoverer 2.3 software (Thermo Fisher Scientific™). SPOT-BioID and SPOT-SupraID enabled the detection of MYB and BSA as Bt-labeled proteins and OVA and BSA as Ad-labeled ones, respectively (Supplementary Fig. 3b, Supplementary Data 1). Thus, the comparative analysis of both results enabled us to detect BSA as a dual-labeled protein. This result indicates the feasibility of SA-Bt and CB[7]-Ad as mutually orthogonal binding systems for detecting target proteins with both chemical labels.

## Two enzymes for orthogonal dual PL with Ad and Bt in live cells

We examined the availability of two engineered enzymes (APEX2 and TurboID) in live cells for dual PL of proteins with Ad and Bt, respectively. Flp-In™ T-REx™ 293 cells expressing APEX2 fused to a lumen-facing site of an ERM protein, such as Sec61B[33] (Sec61B-V5-APEX2), and TurboID fused to a cytosol-facing site of an OMM protein such as TOM20[34] (TOM20-V5-TurboID) were prepared (Fig. 1). Successful expression of both enzymes was confirmed through V5 tag detection via western blotting; here, two different protein bands were detected at 53 kDa and 39 kDa for TOM20-V5-TurboID and Sec61B-V5-APEX2, respectively (Fig. 2a). The enzymes were activated by incubating the cells in media containing Bt and Ad sequentially for 30 min each, followed by 1 min of $H_2O_2$ treatment (Fig. 1). A type of western blot with CB[7]-HRP for detecting Ad-proteins, namely Supra-blot[26], showed a different band pattern from that of western blot with SA-HRP detecting Bt-proteins (Lane 4, Fig. 2b, c). This reflects the functioning of two

different PLs on proteins, one with Ad and the other with Bt. Confocal laser scanning microscopy (CLSM) with AlexaFluor 488-conjugated SA (AF488-SA) and Cy3-CB[7][23] provided selective visualization of Bt- and Ad-proteins, respectively (Fig. 2d). These proteins appeared differently in the cells. Fluorescence signals of AF488-SA and Cy3-CB[7] overlapped well with those of mito-trackers ($r = 0.89$) and blue fluorescence protein-tagged KDEL (BFP-KDEL) ($r = 0.66$), which selectively stained mitochondria and ER where TurboID and APEX2 are expressed, respectively. Neither fluorescent signals from the cells in the CLSM image (Supplementary Fig. 4) nor apparent protein bands in supra blot and western blot analyses (Lane 3, Fig. 2b, c) were observed in control experiments without treatment of the chemical substrates. Therefore, we confirmed the successful activation of TurboID on OMM and APEX2 in ER for dual PL to the enzyme-proximal proteins with Bt and Ad, respectively. Moreover, in the merged image of Cy3-CB[7] and AF488-SA signals (Image 8, Fig. 2d), the existence of cellular regions where fluorescent signals from both are overlapped ($r = 0.51$) indicated that the presence of AdBt-proteins at contact sites of the ER and mitochondria affected by both enzymes.

It is noteworthy that the integrity of MAM proteins appeared unaffected by a 1 min exposure to $H_2O_2$, as confirmed by a known protein band shift assay with the inositol 1,4,5-triphosphate receptor (IP$_3$R), an ER calcium channel at MAM sensitive to reactive oxygen species (ROS)-mediated oxidation (See SI for the details)[35]. In brief, a 30 min exposure of cells to an ROS generator such as $H_2O_2$ or diamide, in conjunction with maleimide conjugated to polyethylene glycol (MAL-PEG, MW 5 kDa), resulted in a significant shift of the IP$_3$R band towards a larger molecular weight region in the gel (Supplementary Fig. 5b3). However, a 1 min exposure of cells to $H_2O_2$—the same conditions used for Ad labeling with APEX in this study—did not induce the IP$_3$R band shift affected by MAL-PEG (Supplementary Fig. 5b2), and was almost identical to the untreated control (Supplementary Fig. 5b1).

## Proteomic identification and analysis of Ad/Bt-labeled proteins using SPOT-SupraID/SPOT-BioID

After enrichment with SA- and CB[7]-beads, Bt- and Ad-proteins were identified by two parallel but distinct proteomic techniques, SPOT-BioID and SPOT-SupraID, respectively. (Figs. 1 and 2e Supplementary Fig. 1). All the LC-MS/MS experiments were conducted in biological triplicates ($R^2 > 0.78$ for SPOT-SupraID and SPOT-BioID, Supplementary Fig. 6) and proteins detected in at least two samples were considered as reliable proteins. As a result, 161 and 712 proteins with a Gene Ontology Cellular Component (GOCC) annotation of ER or mitochondria were identified as Ad- and Bt-labeled proteins, respectively. (Fig. 2f, g, Supplementary Data 2).

Initially, we focused on the Bt-proteins identified by SPOT-BioID and annotated by GOCC as mitochondrial (540) and ER (128) (Fig. 2f, Supplementary Data 2c). Here, we first examined membrane proteins with known transmembrane domains (TMD), namely OMM (36), ERM (51), and Both OMM & ERM (8) (Supplementary Data 3c). Thus, we confirmed that Bt labeling for 54 proteins, including the TOM complex (TOM70 and TOM20), MFN1, and MUL1, occurred on lysine residues exposed to the cytoplasm based on their known topology. This demonstrated the accurate labeling of OMM proximal proteins via TurboID and the identification of labeled proteins by the SA-Bt system. However, owing to the diffusive and promiscuous nature of TurboID-mediated Bt labeling, TurboID alone cannot be used to precisely identify bona fide ER-mito junction localized proteins.

SPOT-SupraID identifies ER proteins affected by APEX2 inside the ER. Therefore, it can provide additional proteomic information for the concise and accurate selection of target proteins at the ER-mito junction, from the broad pool of proteins identified by SPOT-BioID. In this study, among proteins with GOCC annotation as ER or mitochondria, approximately 97% of the Ad-proteins analyzed were ER as well as ER

and mitochondria dual-localized proteins (Fig. 2g, Supplementary Data 2b). Additionally, 63 out of 79 proteins with TMD, including well-known ERM proteins, such as CALX, STT3A, ERLN1, and BAP29, showed the labeling of Ad on the protein tyrosine residues facing the ER lumen where APEX2 was exposed (Supplementary Data 3b). These results also demonstrated the accuracy of APEX2 labeling in our experimental design. Here, we confirmed the applicability of APEX2 with CB[7]-Ad, in conjunction with the other PL system using TurboID with SA-Bt, to accurately identify target proteins without mutual interference.

## Identification of MAM proteins by OrthoID; combinatorial analysis of SPOT-BioID and SPOT-SupraID results

A comparative analysis of the orthogonally obtained results from SPOT-SupraID and SPOT-BioID, as part of the OrthoID process, led to the identification of 21 proteins detected by both systems (Fig. 2h, i, Supplementary Data 2d). The detection of known ERM proteins, such as DNM1L[36], BAP31[37], CISD2[38], CLCC1[39], CKAP4[40], EMD[41], and EMC7[42], at the ER-mito junction demonstrated the high specificity and feasibility of OrthoID for identifying MAM proteins (Supplementary Table 2). Furthermore, in a western blot assay of the AdBt-proteins after sequential enrichment with CB[7]- followed by SA-beads, BAP31 and EMD were clearly detected, while EEA1 (used as a negative control as it was not identified in either SPOT-SupraID or SPOT-BioID) was not (Supplementary Fig. 7). This result demonstrates that proteins at the intersection are indeed labeled with both Ad and Bt due to the concurrent influence of two enzymes at the ER-mito interface.

OMM proteins such as VDAC1[43,44], well-known for their localization at MAM, were also detected as dual-labeled proteins. Given that APEX2 in the ER was fused to Sec61B, a protein translocon localized at the ERM[45], this observation implies that OMM proteins in close proximity to the ERM can also be labeled with Ad by APEX2 as well as Bt by TurboID, owing to the leakage of Ad through Sec61B after being activated by APEX2[46]. However, Ad labeling on cytosol-facing tyrosine motifs of ERM proteins was predominantly limited to Sec61B, indicating that Ad labeling through radical leakage is effective mainly for ERM and OMM proteins that are located close to Sec61B.

Although the proteins that are already reported as MAM proteins in previous studies are predominantly annotated as ERM or ERM-OMM dual-localization (Supplementary Data 2d), to the best of our knowledge, the ones detected as MAM proteins for the first time by OrthoID such as KPYM, MESD, and GANAB, are ER luminal proteins. This distinction might be linked to the localization and orientation of the PL enzymes in OrthoID: APEX2 expressed on ERM points towards the ER lumen, while TurboID on OMM is oriented towards the cytosol. This stands in contrast to other PL-based systems that typically use only one PL enzyme expressed towards the cytosol, allowing OrthoID to target ER proteins more selectively. The three-dimensional rendering of CLSM images and western blots using protein fractions from each organelle revealed the localization of these proteins between the ER and mitochondria (Supplementary Fig. 8b–d, 9), presumably attributed to their functions linked to protein folding and calcium signaling, through which the proteins can move back and forth between the ER lumen and the MAM (Supplementary Table 3)[47–49]. Although OrthoID did not detect certain proteins identified as MAM proteins in other studies[13], such as IP$_3$R, it identifies their interacting partners such as GANAB and KPYM. This suggests that OrthoID is proficient in identifying MAM proteins associated with the ER-mito junctional complex.

Therefore, the combination of a mutually orthogonal dual proteomics approach and binding pair systems, such as SPOT-SupraID and SPOT-BioID, has enabled the identification of proteins located in a specific spatial region, particularly at the ER-mito junction. In this study, OrthoID mainly focuses on an ER-oriented view. Nevertheless, its ability to detect not only the known MAM proteins, but also additional ER luminal proteins that transit the MAM can contribute to

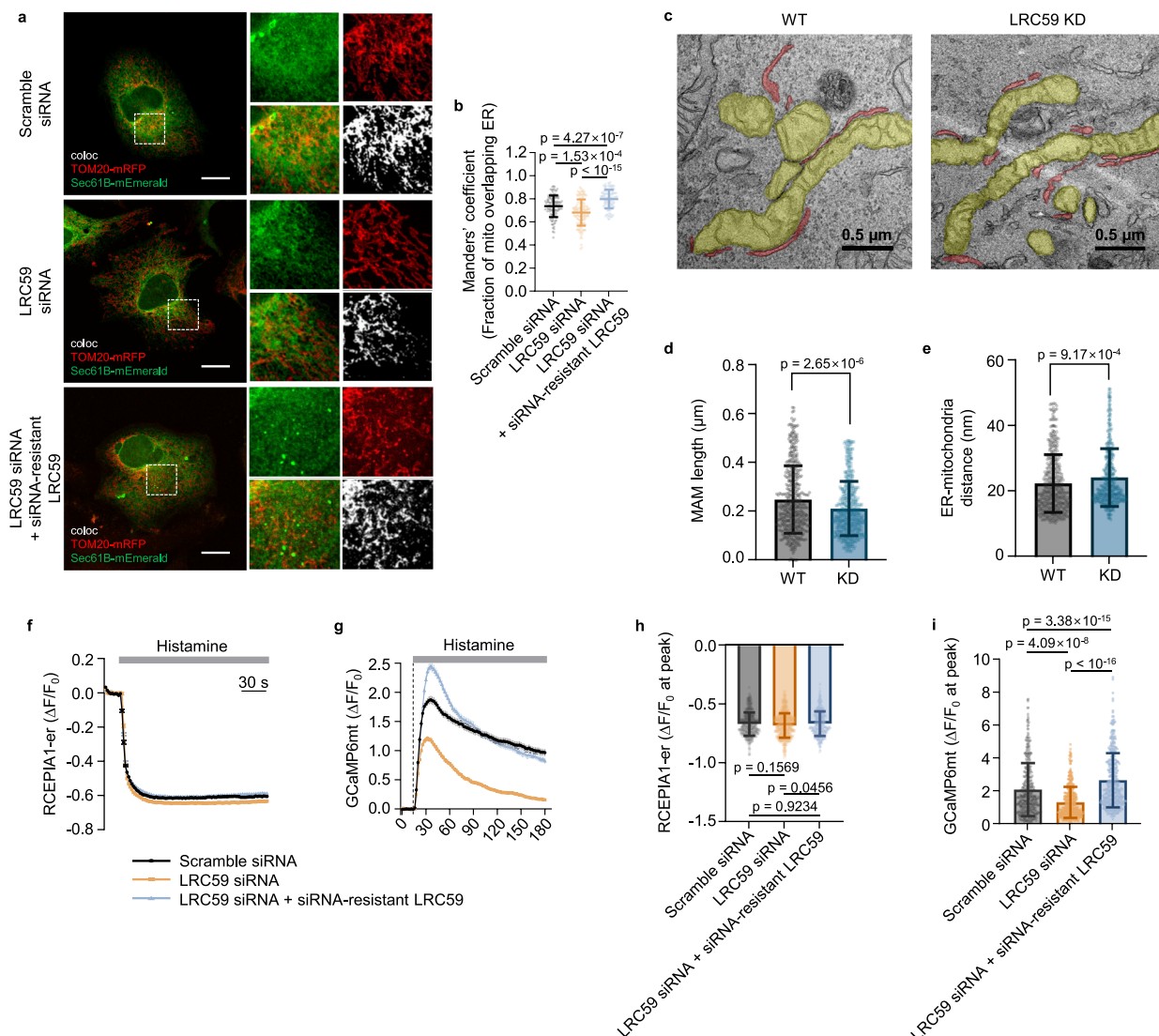

**Fig. 3 | Biological validations of LRC59 protein by ER-mito colocalization assay, TEM analysis, and calcium signaling assay. a** Colocalization assay between the ER and mitochondria with normal, LRC59 knockdown, and re-expressed cells by CLSM with TOM20-mRFP (red) as an OMM marker and Sec61B-mEmerald (green) as an ERM marker. Scale bar: 10 μm. **b** Quantitative analysis of colocalization between ER and mitochondria using Manders' coefficient. (n = 110 cells for scramble siRNA, 133 cells for LRC59 siRNA and 135 cells for LRC59 siRNA+siRNA-resistant LRC59 examined over 3 independent experiments). **c** TEM images of wild type (WT) and LRC59 knockdown (LRC59 KD) cells (yellow: mitochondria, red: ER). **d, e** Statistical analysis of TEM images. **d** Length of ER-mito contact site (MAM) and **e** Distance between ER and mitochondria (the analyses were performed by mapping 486 ER-mito contact sites for the wild type (WT) and 558 ER-mito contact sites for the LRC59 knockdown (KD) in 10–11 cells for each group) **f, g** Changes in fluorescence intensity of calcium indicator protein. **f** RCEPIA1-er and **g** GCaMP6mt after histamine treatment in normal, LRC59 knockdown, and re-expressed cells normalized to the basal signals ($F_0$). **h, i** Bar graphs representing the peak amplitude of $\Delta F/F_0$ from **h** RCEPIA1-er and **i** GCaMP6mt in normal, LRC59 knockdown, and re-expressed cells (n = 501, 512, and 474 cells were used for statistical analysis in normal, LRC59 knockdown, and re-expressed cells with RCEPIA1-er, respectively, and 416, 440 and 395 cells were used for statistical analysis in normal, LRC59 knockdown, and re-expressed cells with GCaMP6mt, respectively.) The error bar in all graphs represents the mean ± SD and the numbers in the graph represent exact p-values. One-way ANOVA and Tukey's post hoc test for multiple comparisons were used to assess statistical differences for **b, h, i**, and two-tailed unpaired Student's t-test for **d, e**. Source data are provided as a Source Data file.

advancing our understanding of protein dynamics at the ER-mitochondria junction and the complex proteomic landscape at the vital cellular interface.

## Involvement of LRC59 in ER-mito contact formation and calcium signaling at MAM

Among the dual-labeled ERM proteins, LRC59 was particularly intriguing as its known topology matches well with the result obtained from OrthoID (Supplementary Fig. 10), yet its function at the ER-mito junction remains elusive. Therefore, we performed a series of biological assays to elucidate the function of LRC59 at MAM. We first verified its localization at the ER-mito junction via subcellular fractionation as well as 3D rendering of CLSM images, visualizing mEmerald fluorescent protein fused to LRC59 appearing between mScarlet fused to Sec61B and blue fluorescent protein (BFP) fused to TOM20, which represent the ER and mitochondria, respectively (Supplementary Fig. 8a, 9). Then, the ER-mito colocalization assay was conducted as previously described[50]. The decrease of ER-mito overlapping in LRC59 knockdown cells by siRNA, as well as its recovery in siRNA-resistant LRC59 re-expressed cells, were subsequently observed. (Fig. 3a, b, Supplementary Fig. 11). By transmission electron microscopy (TEM), we detected an increased distance between the ER and mitochondria, along with a diminished ER-mito contact (MAM) length in LRC59 knockdown cells (Fig. 3c–e). Additionally, we

employed a split green fluorescent protein-based contact site sensor (SPLICS), utilizing one GFP half expressed on ERM and the other on OMM[51]. By CLSM, GFP signals were clearly observed from the control cells as a result of forming a complex of GFP fragments by ER-mito contact. However, these signals notably decreased in LRC59 knock-down cells, followed by signal recovery upon re-expression of LRC59 (Supplementary Fig. 12). These results strongly suggest the involvement of LRC59 in the formation and maintenance of ER-mito contact sites. As one of the main processes regulated by the ER-mito junction is calcium signaling from the ER to mitochondria[52], a calcium signaling assay was additionally performed using fluorescent calcium sensor proteins[53,54]. We detected similar results to those obtained in the colocalization assay, with reduced and restored mitochondrial calcium uptake in LRC59 knockdown and re-expressed cells, respectively (Fig. 3g, i). Contrastingly, there was no difference in calcium release from the ER regardless of LRC59 expression after histamine treatment (Fig. 3f, h). Collectively, these results illustrate a potential role of LRC59 in the interaction between ER and mitochondria, and further demonstrate the accuracy of our developed proteomic approach for the identification of proteins localized between these two organelles.

## Detection of dynamic behaviors of MAM proteins upon induction of mitophagy

This proteomics approach leverages the spatiotemporal feature to analyze the dynamic behaviors of MAM proteins that undergo structural and locational changes during cellular processes such as mitophagy[55–57]. We used SPOT-SupraID and SPOT-BioID to analyze cells treated with carbonyl cyanide 3-chlorophenylhydrazone (CCCP), a parkin-mediated mitophagy inducer[58–61]. Our analysis revealed 27 AdBt-proteins (Fig. 4a–d, Supplementary Data 4d) and identified nine proteins that were exclusive to CCCP-treated cells (Fig. 4e). Further analysis using the Search Tool for the Retrieval of Interacting Genes/Proteins (STRING) revealed that the proteins exclusively detected by our system had a close functional association with well-established mitophagy markers, such as PINK1[62], PRKN[63], MFN2[64], and HUWE1[65], in terms of several aspects of mitophagy, including mitochondrial membrane organization, proteasomal degradation, and regulation of mitophagy in response to mitochondrial depolarization. (Supplementary Fig. 14). As these processes are known to occur between ER and mitochondria (or MAM) to regulate mitochondrial homeostasis via mitophagy, these results demonstrate the capability of our proteomics approach to detect spatiotemporal MAM proteomes associated with mitophagy.

In addition, we detected MTCH2, MIC60, ATD3B, RL40, HCD2, RACK1, and RS3A as MAM proteins during mitophagy (Fig. 4e), whose localization at MAM was previously unknown. Given the functions of the proteins including mitochondrial fission, proteasomal degradation of OMM, autophagosome formation, as well as protein synthesis and import to mitochondria (Supplementary Fig. 15), this spatiotemporal proteomics information provides insights into the timely localized MAM proteins in conjunction with critical steps in the mitophagy pathway. Notably, we found that the mitochondrial inner membrane proteins (ATD3B and MIC60) were detected exclusively after the treatment of CCCP through Ad labeling at intermembrane space (IMS)-exposed tyrosine. This observation strongly suggests that these IMM proteins are susceptible to Ad radicals due to the reduced distance between ER and IMM with OMM in between (Fig. 4f). Given the known roles of ATD3B and MIC60 in stabilizing the IMM and facilitating IMM-OMM contact for regulation of mtDNA replication during mitochondria fission through ER-mito contact (Supplementary Table 4)[66–69], these findings from OrthoID delineate a close contact of not only ER-mito, but also OMM-IMM during mitochondrial fission by the intricate interplay of ER-OMM-IMM. Subsequently, we investigated the proximity between ERM-OMM and OMM-IMM by examining the interactions among specific membrane proteins. Using IP₃R1 and VDAC1 as

markers for ERM and OMM interactions, respectively, and SAM50 and MIC60 (identified in this study) for OMM and IMM interactions, respectively, we conducted a proximity ligation assay (PLA) both with and without CCCP treatment. In CCCP-treated cells, the PLA results revealed an increased count of interactions among these membrane proteins, evident from a greater number of fluorescent PLA dots in CCCP-treated cells than in non-treated (Supplementary Fig. 16). This increase implies a closer membrane proximity, facilitating the labeling of IMM proteins with Ad radicals. These findings not only confirm our earlier proteomic profiling results, but also may provide insights into spatiotemporal cellular events occurring at membrane interfaces.

Eighteen proteins were identified as AdBt-proteins regardless of CCCP treatment (Fig. 4e). Nine of those were analyzed to have different Ad labeling depending on CCCP treatment and based on the comparative analysis of peptide spectrum matches (PSM) identified from the CCCP-treated and non-treated cells (Supplementary Data 5b). ERM and OMM proteins, including ERLN2 (Y85), VDAC1 (Y67, Y173), EMD (Y95), LBR (Y613), TGO1 (Y725, Y969), and CLCC1 (Y46, Y68, Y282) showed additional Ad labeling at the tyrosine residues when mitophagy occurred. This reflects that the residues on the proteins are more accessible to Ad labeling during this cellular process. Thus, it allowed us to suggest the importance of specific protein regions where their structures are sensitively altered during mitophagy. Furthermore, these results clearly demonstrate the potential of OrthoID as a proteomic tool for obtaining a spatiotemporally-resolved snapshot of proteomes associated with a specific and dynamic cellular process, such as mitophagy, which is otherwise highly challenging to achieve.

OrthoID, developed using a mutually orthogonal dual proteomics approach, efficiently identifies MAM protein candidates and provides insights into the dynamic protein behavior during cellular processes like mitophagy. Notably, our experimental design strategically focuses on proteins at a specific spatial region, such as ERM at the ER-mito junction, which probably cannot cover the full spectrum of proteins at ER-mito junctions. In principle, enzyme expression can be modulated in various locations, orientations, and organelles on demand, offering modularity and flexibility of OrthoID. This feature underscores its great potential in pinpointing proteins at different contact sites, including ER-PM, nucleus-vacuole, and mitochondria-lipid droplets, significantly advancing our understanding of organelle interactions. Furthermore, the capability of OrthoID can be expanded by integrating additional orthogonal chemical tools, such as Cu-free click chemistry, along with the non-covalent click system. In principle, this approach can extend to proteins at cellular contact sites for investigating intercellular communication. Thus, OrthoID holds potential as a versatile proteomics tool for in-depth studies of the intricate spatiotemporal interplay between proteins, organelles, and cells in biological communication.

# Methods
## Ethical statement
Animal procedures were approved by the Institutional Animal Care and Use Committee (IACUC) of Pohang University of Science and Technology (POSTECH-2022-0085). All experiments were carried out in accordance with the approved guidelines.

## Reagents, equipment, and antibodies
All the reagents and solvents employed were commercially available and used as supplied without further purification. For protein analysis, the results from sodium dodecyl sulfate-polyacrylamide gel electrophoresis (SDS-PAGE) and western blotting were obtained as images using an ImageQuant LAS500 instrument (GE Healthcare). Proteomic analysis was performed using Q-Exactive Plus (Thermo Fisher Scientific) and Orbitrap Fusion Lumos (Thermo Fisher Scientific) equipped with electrospray ionization (ESI) source and liquid chromatography (LC). The data for proteomics was analyzed with XCalibur software and

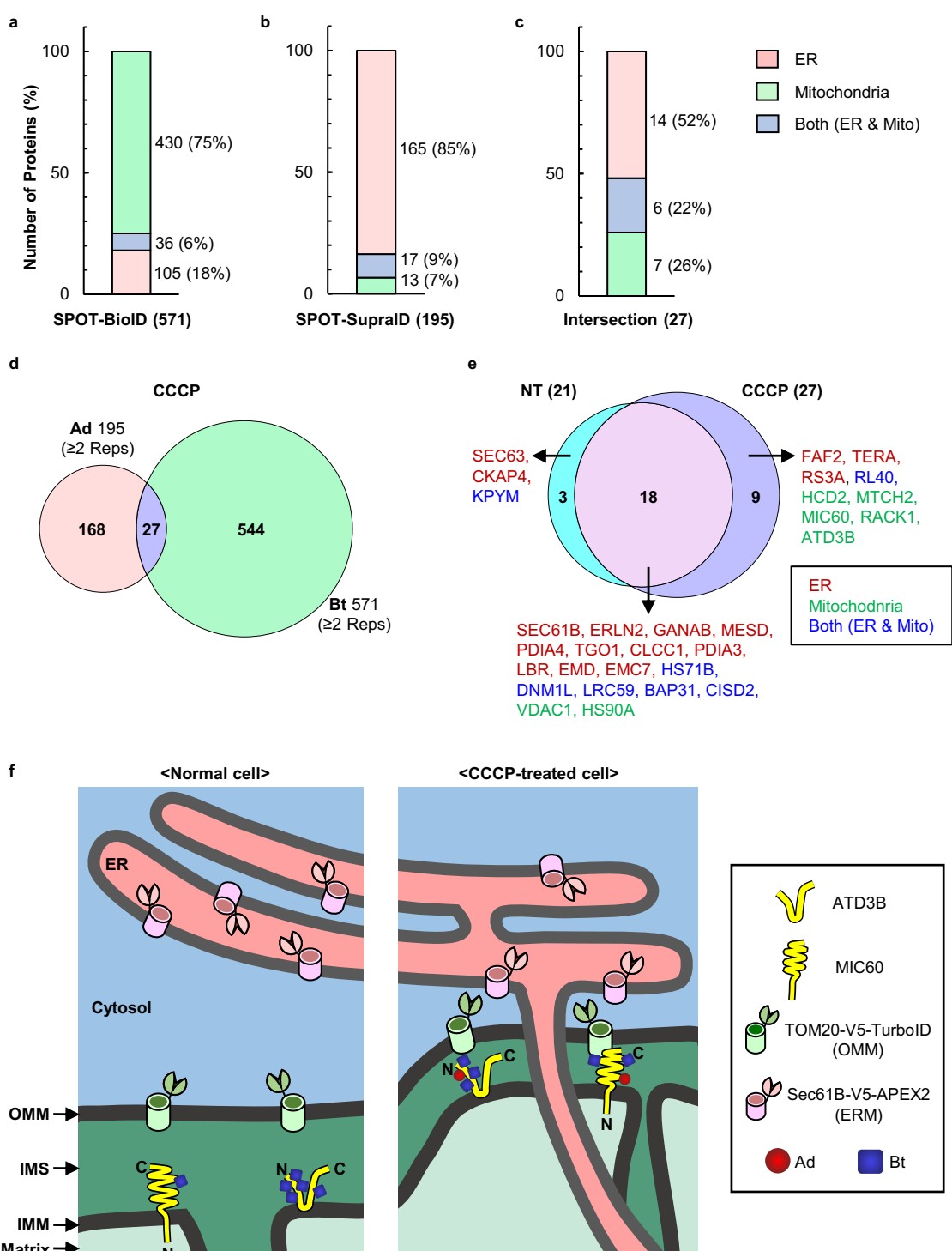

**Fig. 4 | Identification of proteins at ER-mito contact sites after induction of mitophagy. a–c** Bar graphs of the number of total identified proteins from **a** SPOT-BioID, **b** SPOT-SupraID, and **c** intersection of SPOT-BioID and SPOT-SupraID (OrthoID) in CCCP-treated cells with ER and mitochondria annotation. Color represents GOCC annotation of proteins; ER (pink), mitochondria (green), and both

(blue). **d** A Venn diagram of SPOT-SupraID and SPOT-BioID, 27 proteins found at the intersection (OrthoID). **e** A Venn diagram of identified MAM proteins with and without treatment of CCCP. **f** A schematic representation of the topology and labeling sites of IMM proteins (ATD3B, MIC60) exclusively identified by OrthoID in CCCP-treated cells, compared to those in normal cells.

Proteome Discoverer 2.3 (Thermo Fisher Scientific)[1].H-NMR was performed on a Bruker Ascend 850 MHz spectrometer. FT-IR measurements were performed on a Cary 660 FT-IR spectrometer (Agilent Technologies). Mass (MS) analysis for Cy3-CB[7] was performed on a Autoflex Speed LRF MALDI-TOF instrument (Bruker). MS analysis for Ad was performed using a LTQ-XL mass spectrometer (Thermo Fisher Scientific, Inc.) equipped with an electrospray ionization (ESI) source. HPLC was performed on a 1260 Infinity II HPLC system (Agilent Technologies) equipped with a Luna 10 μm C18 LC column (250 mm × 10 mm, 100 Å). Primary and secondary antibodies were used for western blotting and immunofluorescence imaging: anti-BAP31 (Invitrogen, #MA3-002, CC-1, 1:200 dilution), anti-EMD (Proteintech, 10351-1-AP, 1:5000 dilution), anti-LRC59 (Proteintech, #27208-1-AP, 1:5000 dilution), anti-EEA1 (Cell Signaling Technology, #3288, C45B10, 1:500 dilution), anti-V5 (Invitrogen, #R960-25, SV5-Pk1, 1:1000 dilution), Alexa Fluor 488 conjugated Streptavidin (Invitrogen, #S11223, 1:500 dilution), anti-mouse IgG-HRP (Cell Signaling Technology, #7076, 1:1000 dilution), anti-rabbit IgG-HRP (Cell Signaling Technology, #7074, 1:1000 dilution), anti-VDAC1 (Proteintech, 55259-1-AP, 1:200 dilution), anti-IP3R1 (Santa Cruz, sc-271197, E-8, 1:200 dilution / Proteintech, 19962-1-AP, 1: 500 dilution), anti-SAM50 (Santa Cruz, sc-100493, SQ-7, 1:200 dilution), and anti-Mitofilin (Proteintech, 10179-1-AP, 1:200 dilution). Information on the plasmid constructs used in this study is provided in the Supplementary Information as Supplementary Table 1.

## Preparation of Cy3-CB[7][25]

Anhydrous trimethylamine (TEA, 20 μL) was added into a solution of Monoamine Cucurbit[7]uril ((NH$_2$)$_1$-CB[7], 10.0 mg, 7.66 μmol) in anhydrous dimethyl sulfoxide (DMSO 200 μL). The mixture was stirred at 25 °C for 1 h. The NHS-activated Cyanine 3 (5.42 mg, 9.19 μmol) was added to the solution above and stirred for additional 3 h. After completion of the reaction, crude product was purified by HPLC to obtain pure Cy3-CB[7] (5.32 mg, 40%). The following solvent systems were used for HPLC: solvent A (0.1% TFA solution in H$_2$O) and solvent B (0.1% TFA in acetonitrile)[1].H-NMR (850 MHz, D$_2$O): δ = 7.97 (t, 1H), 7.55 (d, 1H), 7.40 (t, 1H), 7.26 (t, 2H), 7.22 (t, 1H), 7.10 (t, 1H), 6.77 (d, 1H), 6.66 (d, 1H), 6.61 (d, 1H), 5.92 (d, 1H), 5.83 (dd, 2H), 5.76 (dd, 2H), 5.69 (d, 1H), 5.67−5.61 (m, 3H), 5.59 (s, 1H), 5.59−5.53 (m, 2H), 5.50 (dd, 2H), 5.46−5.38 (m, 4H), 5.38−5.31 (m, 4H), 5.29 (d, 1H), 5.22 (d, 2H), 5.19 (dd, 2H), 4.52 (dd, 2H), 4.35−4.23 (m, 4H), 4.15 (dd, 3H), 4.06 (d, 2H), 4.00 (dd, 2H), 3.92 (dd, 2H), 3.81 (d, 1H), 3.65−3.55 (m, 2H), 3.53−3.44 (m, 2H), 2.95 (s, 3H), 2.84 (br, 1H), 2.72 (br, 1H), 2.65−2.55 (m, 2H), 2.35−2.20 (m, 2H), 2.09−1.94 (m, 4H), 1.81 (d, 6H), 1.69 (br, 2H), 1.59 (br, 2H), 1.08 (d, 6H). MALDI-MS (m/z): [M-Cl]$^+$ calcd. for C$_{77}$H$_{88}$N$_{31}$O$_{16}$S$^+$, 1734.7; found, [M-Cl+H]$^+$ 1735.7.

## Preparation of CB[7]-beads[27]

Epoxy-Sepharose beads (20 g, 1.4−2.8 mmol mg$^{-1}$ of epoxy functional group density) was added into a sodium hydroxide solution (0.1 M, 5.0 mL) containing Monohydroxy Cucurbit[7]uril ((HO)$_1$CB[7], 20 g, 17 mmol). The heterogeneous solution was mixed by a rotary mixer (20 cm diameter, 5 rpm) at 25 °C for 24 h. The reaction mixture was filtrated by a sintered funnel (pore size: 2 G) and the collected beads were washed with sodium hydroxide solution (0.1 M, 100 mL) followed by water (100 mL) to obtain CB[7]-beads in pure form. The CB[7]-bead solution was stored in water (70 mL at 4 °C). The attachment of (OH)$_1$CB[7] on the beads was characterized by FT-IR signals (ν, cm$^{-1}$) at 1726 (C = O), 1467 (C = N).

## Preparation of phenol-conjugated adamantane (Ad)[23]

4-Hydroxyphenylacetic acid (112 mg, 0.78 mmol), N, N′-diisopropylcarbodiimide (DIC, 98 mg, 0.78 mmol) and N-hydroxysuccinimide (NHS, 90 mg, 0.78 mmol) were added into a solution of N-(2-(2-(2-aminoethoxy)ethoxy)ethyl)adamantan-1-amine (200 mg, 0.70 mmol)

in methylene chloride (MC, 20 mL). Trimethylamine was added to the solution dropwise until it changed to a clear solution and the reaction mixture was stirred for 8 h at room temperature. The crude product was purified by silica column chromatography (eluent, MC:MeOH=9:1). The purified compound was dried under a reduced pressure to give phenol-conjugated adamantane (120 mg, 41%)[1].H-NMR (850 MHz, D$_2$O): δ = 7.19 (d, 2H), 6.87 (d, 2H), 3.69 (br, 2H), 3.62−3.60 (br, 6H), 3.50 (s, 2H), 3.38 (t, 2H), 3.14 (t, 2H), 2.16 (br, 3H), 1.84 (br, 6H), 1.73−1.71 (br, 3H), 1.64−1.61 (br, 3H)[13];C NMR (213 MHz, D2O): δ = 24.8, 28.8, 34.9, 37.9, 38.9, 39.1, 41.6, 57.3, 66.0, 68.7, 69.4, 69.6, 115.8, 126.4, 130.4, 155.3, 175.2, 179.4; ESI-MS (m/z): [M + H]$^+$ calcd. for C$_{24}$H$_{37}$N$_2$O$_4$$^+$, 417.27; found, [M + H]$^+$ 417.42.

## Protein labeling with Ad by APEX2

Cells were prepared in a T75 flask (SPL, #70075). At 80% confluence, cells were treated with doxycycline (5 ng/mL) to induce the enzyme expression. After 24 h, cells were incubated in serum-free DMEM (7.5 mL) with Ad (250 μM) for 30 min. By treating with hydrogen peroxide (H$_2$O$_2$, 750 μL, 10 mM), the labeling was proceeded until the mixture of radical quenchers (2×) including Trolox (10 mM), sodium ascorbate (20 mM), and sodium azide (20 mM) was added to the cells. Cells were rinsed twice with a quenching solution (5 mL, 1×) and harvested for further experiments.

## Protein labeling with Bt by TurboID

Cells were prepared in a T75 flask. At 80% confluence, cells were treated with doxycycline (5 ng/mL) to induce the enzyme expression. After 24 h, the media was exchanged with the one containing Bt (50 μM) to label the proteins with biotin. After 30 min, cells were washed three times with PBS buffer (5 mL) in a T75 flask and harvested for further experiments.

## Western blotting: [1] Antibodies and [2] CB[7]-HRP or SA-HRP

After labelling with Ad and Bt, the harvested cell pellets were lysed with RIPA buffer (50 mM Tris, 150 mM NaCl, 0.1% SDS, 0.5% sodium deoxycholate, 1% Triton X-100) in the presence of a protease inhibitor cocktail (100×, Thermo Fisher Scientific, #78430) for 30 min at 4 °C. The cell lysate was transferred to a microtube and centrifuged at 13,000 × g for 15 min at 4 °C. The supernatant was loaded on a 10% SDS-PAGE gel. After SDS-PAGE, proteins were transferred to a nitrocellulose (NC) membrane, then stained by Ponceau S staining for 1 min. The membrane was blocked with a 5% blocking solution (bovine serum albumin, BSA) or skimmed milk in Tris-buffered saline with 0.1% Tween® 20 (TBST) at 25 °C for 1 h.

[1] Antibodies: The primary antibody was dissolved in 5% blocking solution and incubated with the membrane for 1 h at 25 °C. The membrane was washed five times with 5 mL of TBST. The corresponding secondary antibody was prepared in 5% blocking solution and then incubated with the membrane for 1 h at 25 °C. The membrane was washed five times with 5 mL of TBST. The membrane was developed by treating with Clarity Western ECL Substrate (Bio-Rad) for chemiluminescence imaging.

[2] CB[7]-HRP or SA-HRP: CB[7]-HRP or SA-HRP was dissolved in 5% blocking solution and incubated with the membrane for 1 h at 25 °C. The membrane was washed five times with 5 mL of TBST. The membrane was developed by treating with Clarity Western ECL Substrate (Bio-Rad) for chemiluminescence imaging.

## Fluorescence Cell Imaging: [1] Antibodies and [2] Cy3-CB[7] or Dye-SA

After culturing cells in a confocal dish and labeling with Ad or Bt, the cells were washed three times with PBS and fixed with 4% formaldehyde in PBS for 10 min. After fixation, the cells were permeabilized with 0.5% Triton X-100 in PBS for 15 min and then washed three times with PBS. Cells were blocked with 5% BSA solution in TBST.

[1] Antibodies: The primary antibody was added to the cells for 1 h at 25 °C. The cells were washed five times with TBST. The dye-conjugated corresponding secondary antibody was prepared in 5% BSA in TBST followed by incubation with the cells for 1 h at 25 °C in the dark. The cells were washed five times with TBST. Cells were imaged by a fluorescence microscope.

[2] Cy3-CB[7] or Dye-SA: Cy3-CB[7] (100 nM) or dye-SA (100 nM) in 5% BSA solution was added to the cells for 1 h at 25 °C in the dark. The cells were washed five times with TBST. Cells were imaged by a fluorescence microscope.

## Ad and Bt labeling to bovine serum albumin

Bovine serum albumin (BSA, 1 mg, 14 nmol) was dissolved in PBS (400 μL). Biotin-NHS (341.38 g/mol, 444 μg, 1.3 μmol) dissolved in DMSO (20 μL) was added to the BSA solution in PBS. The mixture was stirred for 15 min at 4 °C. To remove the remaining reagent, the protein solution was purified by a centrifugal filter unit with a 30 kDa molecule weight cut off (MWCO) at $12,000 \times g$ for 10 min. By adding PBS (400 μL), the centrifugal filtration was repeated three more times. The resulting solution was diluted with PBS (final 400 μL) and then treated with Ad (final conc. 250 μM, 100 nmol, 42 μg) and horseradish peroxidase (Sigma #77332, ~150 U/mg, 1 μg). To initiate the radical reaction, $H_2O_2$ (100 mM, 4 μL) was added to the solution. Upon treatment of the quenching solution, a mixture of sodium ascorbate, sodium azide, and trolox (final conc. 10 mM, 10 mM, and 5 mM, respectively), the reaction was terminated. To purify the protein, the resulting solution was filtered with a centrifugal filter unit with a 30 kDa MWCO at $12,000 \times g$ for 10 min. By adding PBS (400 μL), the centrifugal filtration was repeated three more times. Chemical labeling with Ad and Bt were characterized by Supra-blotting with CB[7]-HRP and western blotting with SA-HRP, respectively.

## Ad Labeling to Ovalbumin

Ovalbumin (OVA, 1 mg, 23 nmol) was dissolved in PBS (400 μL). The OVA solution was treated with Ad (final conc. 250 μM, 100 nmol, 42 μg) and horseradish peroxidase (Sigma #77332, ~150 U/mg, 1 μg). To initiate the radical reaction, $H_2O_2$ (100 mM, 4 μL) was added to the solution. Upon treatment of the quenching solution, a mixture of sodium ascorbate, sodium azide, and trolox (final conc. 10 mM, 10 mM, and 5 mM, respectively), the reaction was terminated. To purify the protein, the resulting solution was filtered with a centrifugal filter unit with a 10 kDa MWCO at $12,000 \times g$ for 10 min. By adding PBS (400 μL), the centrifugal filtration was repeated three more times. Chemical labeling with Ad was characterized by Supra-blotting with CB[7]-HRP.

## Bt labeling to myoglobin

Myoglobin (MYB, 1 mg, 59 nmol) was dissolved in PBS (400 μL). Biotin-NHS (341.38 g/mol, 444 μg, 1.3 μmol) dissolved in DMSO (20 μL) was added to the MYB solution in PBS. The mixture was stirred for 15 min at 4 °C. To remove the remaining reagent, the protein solution was purified by a centrifugal filter unit with a 10 kDa MWCO at $12,000 \times g$ for 10 min. By adding PBS (400 μL), the centrifugal filtration was repeated three more times. Chemical labeling with Bt was characterized by western blotting with SA-HRP.

## Supra-blotting and Western blotting for labeled model proteins

The solution of each labeled protein (BSA, OVA, and MYB, 2 μg each) was loaded on a 10% SDS-PAGE gel. After SDS-PAGE, proteins were transferred to a nitrocellulose (NC) membrane, then blocked with a 5% blocking solution (bovine serum albumin (BSA) or skimmed milk in TBST at 25 °C for 1 h. CB[7]-HRP or SA-HRP was dissolved in a 5% blocking solution and incubated with the membrane for 1 h at 25 °C. The membrane was washed five times with 5 mL of TBST. The membrane was developed by treating with Clarity reagent (Bio-Rad) for chemiluminescence imaging.

## Examination of orthogonal enrichment by SPOT-SupraID and SPOT-BioID using model proteins

The labeled model protein mixture (40 μg of BSA, OVA, and MYB) in 2 M urea with 50 mM ammonium bicarbonate (ABC) was treated with 1 M dithiothreitol (DTT, final conc. 10 mM) and incubated for 1 h on the thermomixer (Eppenforf) at 450 rpm, 37 °C. After reduction with DTT, the solution was treated with 400 mM iodoacetamide (IAA, final conc. 40 mM) and incubated for 1 h on the thermomixer at 450 rpm, 37 °C. The solution was then diluted to 1 M urea with 50 mM ABC solution. The protein mixture was treated with 1 M $CaCl_2$ (final conc. 1 mM) followed by trypsin-TPCK (1 mg/mL in water) with the ratio of 20:1 (proteins:trypsin). The resulting solution was incubated overnight on the thermomixer at 450 rpm, 37 °C. After trypsin digestion, the solution was centrifuged at $12,000 \times g$ for 10 min. The supernatant was then incubated with CB[7]-beads (10 μL) or SA-beads (15 μL, Thermo Fisher Scientific, #88817) for 2 h at 25 °C. After incubation with the beads, MS samples were prepared following to the 'Peptide Enrichment for LC-MS/MS (SPOT-ID)' procedure below.

## Cell culture

Flp-In™ T-REx™ 293 (Invitrogen, #R78007), HeLa (Korean Cell Line Bank, #10002) and HEK293T (ATCC, #CRL-3216) cells were cultured in DMEM (Hyclone) supplemented with 10% fetal bovine serum (FBS), L-glutamine (2 mM), penicillin (50 units/mL), and streptomycin (50 μg/ml) at 37 °C under 5% $CO_2$.

## Preparation of Stable Cell Line Expressing Both Sec61B-V5-APEX2 and Tom20-V5-TurboID

The stable cell line expressing Tom20-V5-TurboID was cultured in DMEM (Hyclone) with 10% FBS, L-glutamine (2 mM), penicillin (50 units/mL), and streptomycin (50 μg/ml) at 37 °C under 5% $CO_2$ in a 100 pi dish. Transfection was performed at 60-80% cell confluence with the help of Lipofectamine 2000 (Lipo2000, Invitrogen) by adding a mixture of Lipo2000 (25 μL) and plasmid (5 μg, Sec61B-V5-APEX2) to the cells in a 100 pi dish. After 24 h from transfection, cells were divided into other cell culture dishes (Corning). Geneticin-containing media (100-500 μg/mL) was used as culture media and exchanged every 3–4 days for selecting transfected cells. After the selection for 3–4 weeks, colonies were picked up and transferred to a 24-well plate. Considering the number of cells, cells were transferred to larger plates from time to time to prepare the stocks of stable cells. For the expression test, the stable cell lines were treated with doxycycline (50 ng/mL, Sigma Aldrich) in culture media at 60-80% confluence. After 24 h, cells were then labeled with Ad (250 μM) or Bt (50 μM) and lysed for further analysis.

## Protein labeling with Bt by TurboID and Ad by APEX2: [1] NT cells and [2] CCCP-treated cells

[1] NT cells: Cells were seeded at $1.2 \times 10^7$ in a T75 flask. After 24 h, cells were treated with doxycycline hyclate (5 ng/mL) to induce enzyme expression. After 24 h, cells were treated with Bt (50 μM) to label the proteins by TurboID for 30 min. Then, the media was exchanged to serum-free media containing Ad (250 μM, 7.5 mL) for 30 min. The Ad labeling by APEX2 was initiated upon treatment of $H_2O_2$ (750 μL, 10 mM) for 1 min. To terminate the reaction, a quenching solution (2×, 7.5 mL) including trolox (10 mM), sodium ascorbate (20 mM), and sodium azide (20 mM) were added to the cells and the cells were harvested in a Falcon tube. The harvested cells were washed three times with a quenching solution (1×, 5 mL) for further experiments.

[2] CCCP-treated cells: Cells were seeded at $1.2 \times 10^7$ in a T75 flask. After 24 h, cells were treated with doxycycline hyclate (5 ng/mL) to induce enzyme expression. After 21 h, cells were treated with CCCP (20 μM) for 3 h followed by treatment of Bt (50 uM) in CCCP-containing culture medium to label the proteins by TurboID for 30 min. Then, the media was exchanged to serum-free media

containing Ad (250 μM, 7.5 mL) with CCCP (20 μM) for 30 min. The following steps were the same as those for NT cells.

### Four color imaging of stable cell line expressing Sec61B-APEX2-V5 and Tom20-V5-TurboID

Cells were seeded at $1.5 \times 10^5$ on a confocal dish (SPL, #101350). After 24 h, transfection was performed using Lipo2000 (6 μL) and plasmid DNA encoded with BFP-KDEL (1 μg). Afterward, the cells were treated with doxycycline hyclate (5 ng/mL) to induce enzyme expression. After 24 h, Bt (50 μM) was added to the cells to label the proteins by TurboID for 30 min. Then, the media was exchanged to serum-free media containing Ad (250 μM, 2 mL) for 30 min. The Ad labeling by APEX2 was initiated upon treatment with $H_2O_2$ (200 μL, 10 mM) for 1 min. To terminate the reaction, quenchers (2×) including trolox (10 mM), sodium ascorbate (20 mM), and sodium azide (20 mM) were added to the cells and the cells were washed three times with PBS. Cells were stained with mito-tracker deep-red FM (#M22426, final conc. 100 nM) for 30 min. The residual reagents were washed twice with PBS (2 mL). Cells were fixed with formaldehyde (3.6%) in PBS for 10 min. After washing the cells twice with PBS (2 mL), the cells were permeabilized with Triton X-100 (0.5%) in PBS for 10 min and then washed twice with PBS (2 mL). A BSA solution in TBST (5% of BSA) was added as a blocking agent for 1 h at 25 °C. To stain the proteins labeled with Bt and Ad respectively, Cy3-CB[7] (100 nM) and SA-AF488 (Invitrogen, #S11223, 1:500 dilution) were added to the cells and incubated for 1 h at 25 °C. Before performing fluorescence cell imaging, cells were washed five times with TBST.

### Preparation of protein sample without SDS

After labeling the cells in a T75 flask, cells were lysed with 2% SDS in LC-grade water containing protease inhibitor cocktails (1×) and quenchers (1×). Using a tip sonicator (3 sets, 1 s*15 pulses/set), cells were lysed and centrifuged at $12,000 \times g$ at 25 °C to remove cell debris. The lysate was transferred to a 5 mL Lobind tube (Eppendorf) and mixed with cold acetone (−20 °C) by vortexing and then kept in the freezer (−20 °C) for 2–4 h. The tube was then centrifuged at $13,000 \times g$ for 15 min at 4 °C. After discarding the supernatant, cold acetone (−20 °C) containing 10% TBS buffer was added and incubated for 1–2 h at −20 °C. The resulting solution was centrifuged at $13,000 \times g$ for 15 min at 4 °C and the precipitate was dried for 10 min. Urea (8 M) in ammonium bicarbonate buffer (50 mM, ABC buffer) was used to denature the protein pellet by incubating for 4 h followed by tip sonication (3 sets, 1 s*10 pulses/set). The protein solution was quantified by the bicinchoninic acid assay (BCA assay, Thermo Fisher Scientific).

### Preparation of peptide solution for LC-MS/MS

Proteins in urea solution (8 M) in ABC buffer (50 mM) were treated with DTT (final conc. 10 mM) in LC-grade water for reduction for 1 h at 37 °C. IAA solution (final conc. 40 mM) in LC-grade water was added to the protein solution for alkylation. The solution was diluted from 8 M to 1 M urea solution with ABC buffer (50 mM). $CaCl_2$ (1 M) was added to the diluted solution (final conc. 1 mM) and trypsin-TPCK was added to the solution for the digestion with the ratio of 50:1 (protein:trypsin). The digested protein solution was used for peptide enrichment for proteomic analysis.

### Peptide enrichment for LC-MS/MS (SPOT-ID): [1] CB[7]-beads and [2] SA-beads

[1] CB[7]-beads: CB[7]-beads were added to a peptide solution with a ratio of peptide to beads of 1 mg:50 μL, and acetonitrile (MS-grade, Merck) was added to the mixture (20% v/v). After 2 h of incubation on the rotator at 25 °C, the solution was centrifuged at $1000 \times g$ at 25 °C for 3 min. The supernatant was discarded from the CB[7]-beads and then 30% acetonitrile in ABC buffer (50 mM) was added to the beads for washing (three times). CB[7]-beads were transferred to a size-

exclusion filter (0.45 μm, Merck Millipore, #UFC30HVNB) and then centrifuged at $1000 \times g$ for 1 min to remove the remaining solution. Peptides were eluted from the CB[7]-beads by incubating with adamantane ethylene diamine (Ad-EDA, 4 mM) in ABC buffer (50 mM) for 10 min at 25 °C followed by centrifugation at $1000 \times g$ for 1 min. The eluate was collected into a new 1.5 mL Protein Lobind tube (Eppendorf) and then ABC buffer (50 mM) was re-loaded to the beads to collect the remaining peptides from the CB[7]-beads. The collected peptide solution was cleaned up with a Discovery® DSC-18 SPE Tube (Supelco, #52606-U) and then dried under vacuum (36 °C, 4 h). The dried peptides were re-dissolved in LC-grade water with 0.1% trifluoroacetic acid (TFA) for LC-MS/MS analysis.

[2] SA-beads: SA-beads were added to the peptide solution with the ratio of peptide to beads of 1 mg:75 μL. After 2 h of incubation on the rotator at 25 °C, the beads were collected using a magnet. The beads were washed twice with urea (2 M) in 50 mM ABC (150 μL) and twice with LC grade water (150 μL). After the removal of the supernatant, the SA-beads were heated for 5 min at 60 °C in the eluting solution (100 μL) containing 80% acetonitrile, 0.1% formic acid and 0.2% trifluoroacetic acid in LC-grade water. The elution process was repeated five times and the solutions were collected in another 1.5 mL Protein Lobind tube. The collected solution was dried under a vacuum. The dried peptides were re-dissolved in LC-grade water with 0.1% TFA for LC-MS/MS analysis.

### OrthoID using stable cell line expressing Sec61B-V5-APEX2 and Tom20-V5-TurboID

Bt and Ad co-labeled cells were processed following to the 'preparation of protein sample without SDS' and 'Preparation of peptide solution for LC-MS/MS'. The resulting peptide solutions were processed following to the 'Peptide Enrichment for LC-MS/MS' procedure as biological triplicates.

**NanoLC-MS/MS analysis.** NanoLC-MS/MS analyses were performed on a Q-Exactive Plus mass spectrometer (Thermo Fisher Scientific), an Ultimate 3000 nanoLC pump, and an autosampler (Thermo Fisher Scientific). A fused silica needle (Thermo Fisher Scientific, 2.4 μL, ID 100 μm) packed with Easy Spray column (#ES803A, C18 2 μm spherical fully porous ultrapure silica as reversed phase material, 100 Å pore size, 500 mm metric length, 75 μm metric diameter) was used as an analytical column. The injection volume was 6 μL and the flow rate was 300 nL/min. The mobile phases consisted of (A) 0.1% formic acid in water and (B) 0.1% formic acid in acetonitrile. A gradient program was set with 10% B in 10 min, 10% to 45% B in 90 min, 45% to 90% B in 0.1 min, 90% B in 20 min, 90% to 10% B in 5 min, and 10% B in 5 min. Spray voltages of 1.5 kV were applied. The mass scan ranges were m/z 350–1800 and top 10 precursor ions were selected in each MS scan for subsequent MS/MS scans. The normalized collision energy was set to be 27.

**MS data analysis.** The raw MS data files were analyzed by Proteome Discoverer 2.3 (PD 2.3, Thermo Fisher Scientific) to create peak lists based on the recorded fragmentation spectra. Peptides and proteins were identified by means of automated database searching with SEQUEST against UniprotKB/SwissProt (http://www.uniprot.org) Homo sapiens proteome database for stable cells and Horse, Bovine, and Chick for model proteins, with a precursor mass tolerance of 10 ppm, a fragment ion mass tolerance of 0.02 Da, and trypsin specificity that allows for up to two missed cleavages. Cysteine carbamidomethylation (+57.021 Da) was set as static modification. Methionine oxidation (+15.995 Da) and chemical modifications (Ad (+414.252 Da) on tyrosine and Bt (+226.078 Da) on lysine) were allowed as dynamic modifications. Peptides were considered identified if the peptides were marked as 'Master' in PD 2.3. A reverse decoy database search was conducted to set false discovery rates (FDRs) of less than 1% both at

peptide and protein levels. Proteins were quantified by label-free quantification (LFQ) methods by averaging relative peak intensities of identified peptides. For further proteomic analysis, we focused on proteins identified by SPOT-SupraID and SPOT-BioID that are annotated with the endoplasmic reticulum (ER) and mitochondria, according to their Gene Ontology Cellular Component (GOCC) annotations.

## Immunofluorescence, 3D rendering, and colocalization measurement

For immunofluorescence, HeLa cells were seeded at $1.8 \times 10^5$ in a 12-well plate. Twenty-four hours after plating, cells were co-transfected with LRC59-mEmerald, GANAB-mEmerald, KPYM-mEmerald, or MESD-mEmerald together with an ER marker (Sec61B-mScarlet) and a mitochondrial marker (TOM20-BFP) using Lipo3000 (Invitrogen). Cells were fixed on the following day and 3D rendering of complete z-stack images (confocal microscope, 100× oil objective lens, step of 0.3 μm) were thresholded and constructed by Imaris software (Oxford Instruments). For analysis of ER-mitochondrial colocalization, 24 h after plating, cells were co-transfected with either scramble siRNA, LRC59 siRNA (for LRC59 silencing), or LRC59 siRNA + LRC59-BFP (for LRC59 re-expression) together with an ER marker (Sec61B-mEmerald) and a mitochondrial marker (TOM20-mRFP) at a 1:1 ratio. Cells were fixed after transfection for 72 h. Images were acquired by using a confocal microscope with a 100× oil objective lens. Images were stacked every 0.38 μm and then maximum projected for processing in ImageJ. The analysis of colocalization between two channels (GFP and RFP) involved a series of steps, modified based on the previously reported protocol[50]. First, each channel was processed and segmented to produce masks that isolate specific regions of interest (ROI). Subsequently, the Image Calculator was employed to apply these masks to the original images, resulting in the creation of new images (namely, "GFP_int" and "RFP_int", respectively) that exclusively retained intensities within the masked ROI. Finally, the "JACoP" plugin in ImageJ was used to compute Manders' correlation coefficient between the GFP_int and RFP_int channels, based on the intensity-preserved images.

## Calcium signaling assay

HeLa cells were seeded at $1.8 \times 10^5$ in a 24-well plate and transfected with plasmids (GCaMP6mt or RCEPIA1-er, i.e., mitochondria-targeted and ER-targeted genetically encoded calcium indicators, respectively) on the following day using Lipo3000 (Invitrogen). The media were changed to DMEM complete 4 h after transfection. Live calcium imaging was performed 72 h after transfection. Cells were imaged in an extracellular calcium buffer (Hank's buffered salt solution (HBSS) containing NaCl 150 mM, KCl 4 mM, $CaCl_2$ 2 mM, $MgCl_2$ 1 mM, D-glucose 5.6 mM, and HEPES pH 7.4 25 mM). Calcium release was triggered by 200 μM histamine (Sigma Aldrich). Time-lapse images were conducted every 2 s under an inverted microscope (OLYMPUS, model IX71) with a UPLSAPO 20×/0.75 NA objective and a CCD camera (model C9100-13, Hamamatsu Photonics). The fluorescence intensities were processed, and background was subtracted by Cellsense software (Olympus). Data was displayed as $\Delta F/F_0$ or $(F - F_0)/F_0$, where $F_0$ is the baseline fluorescence signal of GCaMP6mt or RCEPIA1-er averaged over a 20 s period prior to histamine stimulation, and F is the actual intensity at each recorded time point.

## SPLICS$_s$ intensity measurement

HeLa cells were seeded in a 12-well plate at the density of $1.8 \times 10^5$ cells per mL and transfected with either scramble or siLRC59 or siLRC59+-siRNA-resistant LRC59 (LRC59-BFP) on the following day using Lipofectamine 3000 (Invitrogen). Cells were maintained in DMEM (Hyclone) supplemented with 10% (v/v) fetal bovine serum (Thermo Scientific) and 1% penicillin/streptomycin (DMEM complete). To avoid the poorly reversible characteristic of GFP fragments once

reconstitution, bicistronic plasmid expressing equimolar amounts of ER short and mitochondrial GFP fragments (SPLICS Mt-ER Short P2A, Addgene, plasmid #164108) were additionally transfected for 24 h before fixing. Cells were fixed 72 h after transfection. Single plane images were taken by using a 60x oil immersion confocal microscope (OLYMPUS, model IX71) and the GFP intensity was measured by ImageJ using Mean gray value parameter.

## Proximity ligation assay (PLA)

Spatial associations between VDAC1 and $IP_3R1$ at ER-mitochondria interfaces, as well as between SAM50 and MIC60 across the outer and inner mitochondrial membranes, were examined through Duolink® in situ proximity ligation assay (PLA), as per the supplied protocol. After incubation with CCCP for 3 h, HeLa cells underwent fixation in 4% paraformaldehyde, permeabilization with 0.2% Triton X-100, and blocking, followed by overnight incubation at 4 °C with specific primary antibodies: rabbit anti-VDAC1 (Proteintech, 55259-1-AP), mouse anti-$IP_3R1$ (Santa Cruz, sc-271197), mouse anti-SAM50 (Santa Cruz, sc-100493), and rabbit anti-Mitofilin (Proteintech, 10179-1-AP). Subsequent steps involved 1 h PLA probe incubation, 30 min ligation, and 2 h amplification at 37 °C. Hoechst staining facilitated nuclear visualization. Confocal imaging was performed using an Olympus FV3000 microscope with an UPLSAPO 100×/1.35 NA oil objective. Texas red-labeled probes (Sigma Aldrich, DUO92008) identified PLA signals denoting proximities between ER-mitochondria and OMM-IMM. Quantitative analysis of PLA red dots per cell was conducted using Imaris software.

## Animals

12-week-old ICR mice used for subcellular fractionation were purchased from Hyochang Science (Daegu, South Korea). There is no need for housing as all the mice were sacrificed immediately after purchase.

## Subcellular fractionation

The detailed protocol has been described previously[70]. Briefly, whole brain from five 12-week-old ICR mice (Hyochang Science, Daegu, South Korea) was collected and homogenated in IB-1 buffer (225 mM mannitol, 75 mM sucrose, 30 mM Tris-HCl pH 7.4, 0.5 mM EGTA, and 0.5% BSA). The homogenate was washed three times by centrifugation at $700 \times g$ for 5 min to remove nuclei and unbroken cells. The pellet of crude mitochondria and the supernatant containing ER and cytosol were simultaneously obtained after centrifugation at $10,000 \times g$ for 10 min. The crude mitochondrial pellet was then resuspended in MRB buffer (250 mM mannitol, 5 mM HEPES pH 7.4, and 0.5 mM EGTA) and layered on top of a 30% Percoll gradient. After centrifugation at $95,000 \times g$ for 1 h, the heavy fraction (pure mitochondria) and light fraction (MAM) were recovered and washed in MRB buffer by centrifugation at $6,300 \times g$ for 10 min. The pellet of pure mitochondria was resuspended in MRB buffer while the supernatant containing MAM was pelleted by centrifugation at $100,000 \times g$ for 1 h. The MAM pellet was lastly resuspended in MRB buffer. To separate ER and cytosol fractions, the supernatant obtained above was subjected to centrifugation at $20,000 \times g$ for 30 min and further at $100,000 \times g$ for 1 h to give a supernatant (cytosol) and a pellet (ER). The ER pellet was lastly resuspended in IB-1 buffer. A BCA assay was carried out to determine protein concentrations. Around 80 μg of proteins per well were separated on 7–15% SDS-PAGE gels and followed by immunoblotting.

## Protein band shift assay

The experiments were carried out following the previous report[35]. Specifically, HeLa cells seeded at $8 \times 10^6$ in a 100 mm dish were treated with $H_2O_2$ (1 mM, 1 min) or diamide (4 mM, 30 min). The reaction was quenched with 20% ice-cold trichloroacetic acid (TCA) immediately after removal of the reagent. The collected protein precipitates were centrifuged at 4 °C, the supernatant was decanted and the protein

pellets were dissolved in denaturing buffer (DB, 200 mM Tris-HCl (pH 8.5), 10 mM EDTA, 0.5% SDS, 6 M urea). The protein solution was then treated with IAA (10 mM) for 45 min, and reprecipitated with 10% ice-cold TCA. After precipitation, the protein pellets were redissolved in DB containing 10 mM DTT for 45 min. The proteins were re-precipitated in 10% ice-cold TCA, resuspended in DB and the pH adjusted to 7.0 with sodium hydroxide (NaOH). After the BCA assay, 15 μg of protein was incubated with 0.5 mM methoxypolyethylene glycol maleimide (Sigma Aldrich) for 1 h. Protein samples were loaded onto a 5% SDS-PAGE gel followed by western blot using anti-IP$_3$R1 antibody (Protein tech, 19962-1-AP).

## TEM analysis
For LRC59 knockdown cells, HeLa cells were transducted with lenti-virus for 3 days prepared by following method: HEK293T cells were transfected with the packaging plasmids psPAX2 (Addgene #12260), pMD2.G (Addgene #12259), pLL3.7-GFP-LRC59-shRNA or pLL3.7-GFP as a control. The cells were prefixed with 2% paraformaldehyde and 2.5% glutaraldehyde in 0.15 M sodium cacodylate buffer (pH 7.4) for 1 h. After washing three times in 0.15 M cacodylate buffer, cells were postfixed in 2% osmium tetroxide (EMS, Hatfield, PA, USA, #19150) with 1.5% ferrocyanide in 0.1 M sodium cacodylate buffer on ice for 1 h. After washing three times in ice-cold ddH$_2$O, they were en bloc stained with 1% uranyl acetate overnight at 4 °C in the dark. The cells were then washed three times in ddH$_2$O for 5 min. For embedding, samples were dehydrated through a graded ethanol series (30, 50, 70, 80, 90, 100, and 100% for 20 min each, all cooled to 4 °C) followed by epoxy resin infiltration by immersion into 3:1, 1:1, and 1:3 mixtures of ethanol and Epon 812 resin (EMS) at room temperature for 1 h. The infiltrated samples were then incubated in pure resin overnight and placed in an inverted capsule on a confocal dish in a pre-warmed oven (70 °C) for 48 h. After trimming the epoxy resin-embedded cells, ultrathin serial sections with 70 nm thickness were cut using an ultramicrotome (Leica, EM UC7) and mounted on 0.25% formvar coated on hole grids. To enhance the electron density, the sections were stained with UranylLess (EMS, #22409) and 3% lead citrate (EMS, #22410). TEM images were acquired using a Tecnai G2 20 (Thermo Fisher Scientific) at 120 kV with a US1000XP CCD detector (Gatan, Pleasanton, CA, USA). For the analysis of electron microscopy (EM) images, we employed the brush selection tool in Microscopy Image Browser (MIB)[71]. This facilitated the selective identification of mitochondria and MAM regions, which were then converted into masks for object recognition. Quantification of MAM was achieved through two distinct methods: firstly, by calculating the MAM length, and secondly, by measuring the distance between ER and mitochondrial membranes.

## Statistics and reproducibility
LC-MS/MS experiments using stable cells were performed in biological triplicates, and bio-validation experiments including calcium signaling, split-GFP, PLA, and ER-mito colocalization assays were performed three times independently, with at least 80 cells used for statistical analysis. For electron microscope image analysis, 10 to 11 cells were used for each group and more than 450 ER-mito contact sites were analyzed for statistics. No statistical method was used to predetermine sample size and no data were excluded from the analyses. Quantitative data were presented as the mean and Standard Deviation (SD), unless otherwise stated. Statistical analyses were performed by GraphPad Prism 9.0. To compare the means of two separate groups, a two-tailed independent-samples t-test was used. If the means of more than two groups needed to be compared, a one-way analysis of variance (ANOVA) was applied, followed by Tukey's post-hoc test. A significance level of 0.05 was chosen as the benchmark.

## Reporting summary
Further information on research design is available in the Nature Portfolio Reporting Summary linked to this article.

## Data availability
The MS data reported in this study have been deposited in the ProteomeXchange Consortium via the jPOST partner repository under accession code "PXD040926" for ProteomeXchange and "JPST002091" for jPOST. The processed proteomics data are provided as Supplementary Data files. Source data are provided with this paper.

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

## Acknowledgements

This work was supported by the Institute for Basic Science [IBS-R007-D1] for K.K., the National Research Foundation of Korea [NRF-2023-00211758] for K.M.P., [NRF-2022R1A2B5B03001658] for H.W.R., [IBS-R008-D1, NRF-2021R1A2C2009336] for J.S.K., [NRF K-Brain Project RS-2023-00265581, Innovation Research Center RS-2023-00260454] for S.K.P., and [NRF K-Brain Project RS-2023-00262332] for J.Y.M.

## Author contributions

H.W.R., K.M.P., and K.K. conceived the idea. A.L. and G.S. contributed equally to the work. A.L., G.S., K.M.P., and K.K. wrote the manuscript with input from all authors. A.L. and G.S. performed all experiments, unless otherwise stated. A.L., G.S., S.S., S.-Y.L., J.S.K., H.W.R., and K.M.P. performed MS spectrometry and data analysis. A.L., G.S., J.S., performed the sample preparation for protein enrichment. A.L., G.S., and P.P.S. prepared the materials, A.L., G.S., T.T.M.N., T.D.N., S.K.P., and K.M.P. performed the bio-validation experiments, A.L., G.S., and P.P.S. prepared the materials, A.L., G.S., T.T.M.N., T.D.N., S.K.P., and K.M.P. performed the bio-validation experiments, A.L., T.T.M.N., T.D.N., I.K., J.Y.M., and K.M.P. performed the TEM analysis. All authors discussed the results and approved the final version of the manuscript.

## Competing interests

The authors declare no competing interests.
