## [Peer Review File · Nature Communications]

REVIEWER COMMENTS

Reviewer #1 (Remarks to the Author):

This is a beautiful paper lead by Kimoon Kim on the development of a pair of mutually orthogonal recognition modules (Bt-SA and CB7-Ad) which allows for the profiling of dynamic proteomes by dual proximity labelling and identification of proteins at the junction of the endoplasmic reticulum and mitochondria. Very significantly, the work indentified new proteins associated with the ER-mitochondria contact including LRC59. Furthermore, the authors were able to observe the behavior of MAM proteins after mitophagy induction including the identification of 7 new proteins whose localization at MAM was not know previously. The work is very carefully performed and the methodology and interpretations of the data are sound. Overall, the work establishes that OrthoID - is a powerful proteomics tool to study the contact sites between organelles and their dynamic processes. The work is a true tour-de-force and is certain to be impactful to the chemical and biological research communities.

It is my pleasure to strongly recommend publication in Nature Communications after suitable correction of some typographical issues listed below.

Page 3, line 56 - "high spatiality" is somewhat unclear, please rephrase.

Page 4, line 103 - "highly spatiotemporal proteins" is somewhat unclear, please rephrase.

Page 6, line 213-4: "highly spatial proteins localized" - please rephrase.

Reviewer #2 (Remarks to the Author):

Lee et al have developed a new strategy for the proteomic detection of membrane contact site (MCS) proteins. It is based on TurboID and APEX2 and they have called it OrthoID. A synthetic host molecule, CB7, can form a complex with adamantylammonium to isolate proteins through the use of an APEX-based proximity labeling. The study is of interest to the field. However, better integration into what is currently known about MAM markers would be helpful for the reader. APEX labeling must control for the integrity of the MAM and for the absence of MAM modification, a phenomenon that could increase the number of proteins detected artificially.

Specific points:

1. The discussion of known MAM proteins on both the ER and mitochondria that were discovered through the different methods used in the screens is currently insufficient. The authors should point out which markers were detected with which method.
2. APEX labeling uses H₂O₂. The authors must control for the absence of MAM-modification since this membrane contact is a site of known ROS flux and modification. This reviewer is concerned that the H₂O₂ used could alter MAM properties and lead to false positives.
3. The discussion about LRC59 is currently very short. What is known about the sequence or topology of this protein?

Reviewer #3 (Remarks to the Author):

In this manuscript, Lee and Sung et al. reported a method OrthoID combining distinct proximity labeling enzymes (TurboID and APEX2) and orthogonal enrichment strategies (biotinylation and adamantylation) to achieve duplex-labeling. A highly similar strategy is reported by another group in bioRxiv (<https://www.biorxiv.org/content/10.1101/2023.02.07.527548v1>). In this study by Lee and Sung et al., the authors used the ER-mitochondrion contact site to demonstrate OrthoID's applicability for mapping the subcellular proteome at a high spatial and temporal resolution and identified novel proteins, such as LRC59, localized at the ER-mito interface. Moreover, the authors profiled how mitophagy changes the proteome of ER-mito contact site. The method seems solid and is of interest to the field of spatially-resolved proteomics but the following caveats need to be addressed:

- The current proteomic design could identify proteins present at the ER-mito contact site but not necessarily enriched there. For instance, proteins uniformly present on the ER membrane with high abundances would be detected. These proteins are not necessarily related to the formation or function of the ER-mito contact site. A ratiometric strategy previously described by the Ting lab would eliminate this kind of false positives.

- Only ER transmembrane proteins could be detected by this method, introducing false negatives in the ER-mito contact site proteome. For instance, lipid-anchored proteins or cytosolic proteins enriched at the ER-mito contact site would not be captured and identified.

- There are previous publications describing the proteome of the ER-mito contact site using different approaches. The authors should systematically and quantitatively compare the proteomic results. Are

previously described ER-mito contact proteins detected in the current study? If there are discrepancies, the authors should thoroughly discuss potential reasons.

- Like any other proximity labeling or general pull-down experiments, negative controls are crucial for interpreting the specificity of the method and results. The proteomic design and mass spectrometry results are not fully clear here. The authors should describe whether non-labeling and single-labeling (biotinylation only and adamantylation only) controls were done and how they were used in data analysis and interpretation.

- The functional studies are preliminary. The study of LRC59 is not fully convincing, particularly for the confocal imaging part. The authors should use electron microscopy and/or super-resolution microscopy to better examine whether LRC59 influences ER-mito contact formation or not.

- The mitophagy data is largely limited to proteomic profiling. Expression validation and/or functional investigation are highly encouraged. The authors made a few claims on ATD3B and MIC60, which requires further evidence.

Reviewer #1 (Remarks to the Author):

This is a beautiful paper lead by Kimoon Kim on the development of a pair of mutually orthogonal recognition modules (Bt-SA and CB7-Ad) which allows for the profiling of dynamic proteomes by dual proximity labelling and identification of proteins at the junction of the endoplasmic reticulum and mitochondria. Very significantly, the work indentified new proteins associated with the ER-mitochondria contact including LRC59. Furthermore, the authors were able to observe the behavior of MAM proteins after mitophagy induction including the identification of 7 new proteins whose localization at MAM was not know previously. The work is very carefully performed and the methodology and interpretations of the data are sound. Overall, the work establishes that OrthoID - is a powerful proteomics tool to study the contact sites between organelles and their dynamic processes. The work is a true tour-de-force and is certain to be impactful to the chemical and biological research communities. It is my pleasure to strongly recommend publication in Nature Communications after suitable correction of some typographical issues listed below.

Page 3, line 56 - "high spatiality" is somewhat unclear, please rephrase.

Page 4, line 103 - "highly spatiotemporal proteins" is somewhat unclear, please rephrase.

Page 6, line 213-4: "highly spatial proteins localized" - please rephrase.

→ We appreciate the reviewer's kind support of our manuscript and suggestions regarding the clarification of specific phrases within the manuscript. Following the reviewer's suggestion, we have now reworded the phrases as follows:

Page 3, line 56 - The term "high spatiality" has been replaced by "proteins localized in specific subcellular regions".

Page 4, line 103 - The term "highly spatiotemporal proteins" has been revised to "proteins that are present in specific subcellular compartments at specific times".

Page 6, line 213-4 - The term "highly spatial proteins localized" has been changed to "proteins located in a specific spatial region, particularly at the ER-mito junction".

Reviewer #2 (Remarks to the Author):

Lee et al have developed a new strategy for the proteomic detection of membrane contact site (MCS) proteins. It is based on TurboID and APEX2 and they have called it OrthoID. A synthetic host molecule, CB7, can form a complex with adamantylammonium to isolate proteins through the use of an APEX-based proximity labeling. The study is of interest to the field. However, better integration into what is currently known about MAM markers would be helpful for the reader. APEX labeling must control for the integrity of the MAM and for the absence of MAM modification, a phenomenon that could increase the number of proteins detected artificially.

→ We thank the reviewer for his/her appreciation of our work and critical comments. Following the reviewer's suggestions, we have now revised the manuscript. We believe that these changes have greatly contributed to improving our manuscript, with which we hope the reviewer will be satisfied.

Specific points:

1. The discussion of known MAM proteins on both the ER and mitochondria that were discovered through the different methods used in the screens is currently insufficient. The authors should point out which markers were detected with which method.

→ Following the reviewer's thoughtful suggestion, we have now included Table S2 in the revised supplementary information, indicating the markers, their functions and methods to identify them.

Table S2. Detection methods and functions of known MAM proteins identified by OrthoID

***References in research article (proximity labeling-based method)**

A: APEX coupled with biochemical fractionation^{S1}, B: APEX2 at the ERM and OMM towards the cytosol^{S2}

C: Contact-ID (split pBirA)^{S3}, D: Split-TurboID^{S4}

****References in review article: E^{S5}, F^{S6}, G^{S7}, H^{S8}, I^{S9}, J^{S10}**

Accession	Gene Name	*References in research article (PL-based method)	**References in review article	Confirmation method	MAM-related function
Q96S66	CLCC1	A, C, D	J	Fluorescence microscopy ^{S11}	Interacting partner of mitochondria microprotein (PIGBOS) to regulate unfolded protein response
Q9UGP8	SEC63	A, C			
P60468	SC61B	A, C			*ERM Translocon conjugated to APEX2
O94905	ERLN2		J	Organelle fractionation ^{S12-13}	Cholesterol homeostasis, initiation of autophagy ^{S14}

P13667	PDIA4	A			
P30101	PDIA3		G	Organelle fractionation ^{S15}	Regulation of calcium homeostasis ^{S16-17}
Q14739	LBR	C, D			
Q07065	CKAP4	C	J	Immunoprecipitation ^{S18}	Regulation of mitochondria function, calcium influx
P50402	EMD	C, D	F	Organelle fractionation ^{S19}	Spacer/linker at MAM
Q9NPA0	EMC7	C	F		Phospholipid transfer from ER to mitochondria ^{S20}
O00429	DNM1L	D	E, F, G, I, J	Fluorescence microscopy ^{S21-23}	Mitochondria fission
Q96AG4	LRC59	A, C		Fluorescence microscopy, Organelle fractionation (in this study)	Involved in the formation of ER-mito contact (in this study)
P51572	BAP31	A, D	F, H	Fluorescence microscopy ^{S24}	Apoptosis signaling
Q8N5K1	CISD2	A, C			Maintenance of calcium homeostasis ^{S25}
P21796	VDAC1	B	E, F, H, I, J	Organelle fractionation ^{S26}	Calcium signaling

2. APEX labeling uses H₂O₂. The authors must control for the absence of MAM-modification since this membrane contact is a site of known ROS flux and modification. This reviewer is concerned that the H₂O₂ used could alter MAM properties and lead to false positives.

→ In response to the reviewer's concern, we performed an extra experiment, a protein band shift assay utilizing the inositol 1,4,5-triphosphate receptor (IP₃R) as a model MAM protein (See below, **Figure S5**). IP₃R is known to contain cysteine that is sensitive to reactive oxygen species (ROS)-mediated oxidation. This oxidation can affect the redox state of thiols in IP₃R, potentially impacting MAM integrity. (See ref. 34 in the revised manuscript). We employed maleimide conjugated polyethylene glycol (MAL-PEG, MW 5 kDa) to detect oxidized thiols in IP₃R. The results revealed that a 30 min treatment with an ROS generator such as hydrogen peroxide or diamide, in conjunction with polyethylene glycol maleimide (MAL-PEG, MW 5 kDa) caused a significant shift of the IP₃R band towards a larger molecular weight region in the gel (**Figure S5B3**). However, a 1 min exposure to hydrogen peroxide, the same condition used for Ad labeling with APEX in this work, did not produce the IP₃R band shift affected by MAL-PEG (**Figure S5B2**), and was almost identical to the untreated control (**Figure S5B1**). This indicates that a brief (1 min) exposure of the cells to hydrogen peroxide in our Ad labeling protocol does not

induce apparent modification of IP₃R, suggesting that our Ad labeling conditions are unlikely to alter MAM properties. We have updated both the manuscript and supplementary information with detailed experimental descriptions to reflect these results. *“It is noteworthy that the integrity of MAM proteins appeared unaffected by a 1 min exposure to H₂O₂, as confirmed by a known protein band shift assay with the inositol 1,4,5-triphosphate receptor (IP₃R), an ER calcium channel at MAM sensitive to reactive oxygen species (ROS)-mediated oxidation (See SI for the experimental details). In brief, a 30 min exposure of cells to an ROS generator such as H₂O₂ or diamide, in conjunction with maleimide conjugated to polyethylene glycol (MAL-PEG, MW 5 kDa), resulted in a significant shift of the IP₃R band towards a larger molecular weight region in the gel (Figure S5B3). However, a 1 min exposure of cells to H₂O₂—the same conditions used for Ad labeling with APEX in this study—did not induce the IP₃R band shift affected by MAL-PEG (Figure S5B2), and was almost identical to the untreated control (Figure S5B1).”*

Figure S5. Protein band shift assay of IP₃R protein.

(A) Schematic description of protein shift assay of IP₃R. IAA represents iodoacetamide and PEG-MAL represents methoxypolyethylene glycol-maleimide (MW: 5k).

(B) Western blot using anti-IP₃R1 antibody. 1: non-treated, 2: H₂O₂ treated (1 mM, 1 min), 3: diamide treated (4 mM, 30 min)

3. The discussion about LRC59 is currently very short. What is known about the sequence or topology of this protein?

→ Following the reviewer’s comment, we have now added the known information on the sequence and topology of LRC59 to **Figure S10** (see below) and discussed it accordingly in the revised manuscript as follows. *“LRC59 was particularly intriguing as its known topology matches well with the result obtained from OrthoID (Figure S10), yet its function at the ER-mito junction remains elusive. Therefore, we performed a series of biological assays ~”*

A

10	20	30	40	50	
MTKAGSKGGN	LRDKLDGNEL	DLSLSDLNEV	PVKELAALPK	ATILDLSCNK	
60	70	80	90	100	
LTTLPSTDFCG	LTHLVKLDLS	KNKLQQLPAD	FGRLVNLQHL	DLLNNKLVTL	Cytosol
110	120	130	140	150	
PVSFAQLKNL	KWLDLKDNP	DPVLAKVAGD	CLDEKQCKQC	ANKVLQHMKA	
160	170	180	190	200	
VQADQERERQ	RRLEVEREAE	KKREAKQRAK	EAQERELRKR	EKAEEKERRR	
210	220	230	240	250	ER membrane
KEYDALKAAK	REQEKKPKKE	ANQAPKSKSG	SRPRKPPPRK	HTRSWAVLKL	
260	270	280	290	300	ER lumen
LLLLLLFGVA	GGLVACRVTE	LQQQPLCTSV	NTIYDNAVQG	LRRHEILQWV	
LQTDSQQ					

Figure S10. Labeled peptide sequence and topology of LRC59.

(A) Known sequence of LRC59 from Uniprot database and Ad/Bt-labeled sites by OrthoID (K (lysine) with Bt (blue) and Y (tyrosine) with Ad (red)).

(B) Schematic description of labeling sites-based topology of LRC59 by OrthoID.

Reviewer #3 (Remarks to the Author):

In this manuscript, Lee and Sung et al. reported a method OrthoID combining distinct proximity labeling enzymes (TurboID and APEX2) and orthogonal enrichment strategies (biotinylation and adamantinylation) to achieve duplex-labeling. A highly similar strategy is reported by another group in bioRxiv (<https://www.biorxiv.org/content/10.1101/2023.02.07.527548v1>). In this study by Lee and Sung et al., the authors used the ER-mitochondrion contact site to demonstrate OrthoID's applicability for mapping the subcellular proteome at a high spatial and temporal resolution and identified novel proteins, such as LRC59, localized at the ER-mito interface. Moreover, the authors profiled how mitophagy changes the proteome of ER-mito contact site. The method seems solid and is of interest to the field of spatially-resolved proteomics but the following caveats need to be addressed:

→ We would like to express our gratitude for the reviewer's constructive comments. In the light of the reviewer's comments listed below, we have now revised our manuscript and supplementary information, accordingly. We hope that the reviewer will find the revised manuscript satisfactory.

1. The current proteomic design could identify proteins present at the ER-mito contact site but not necessarily enriched there. For instance, proteins uniformly present on the ER membrane with high abundances would be detected. These proteins are not necessarily related to the formation or function of the ER-mito contact site. A **ratiometric strategy** previously described by the Ting lab would eliminate this kind of false positives.

→ In light of the reviewer's comments, we re-evaluated our data employing a ratiometric strategy using label free quantification (LFQ, see ref.7). This involved the utilization of SPOT-ID techniques via tandem mass spectrometric analysis, enabling highly reliable detection of peptides with increased mass values on lysine (K + 226.078 Da) through Bt labeling (SPOT-BioID) and tyrosine (Y + 414.252 Da) through Ad labeling (SPOT-SupraID).

In brief, we performed an additional experiment utilizing APEX2-NES cells expressing APEX2 in cytosol as a control to cells with Sec61B-APEX2 on the ERM facing the ER lumen which we used in this work. Ratiometric analysis on the results from both cell types led to the removal of 11 proteins from the initial list of 161 Ad-proteins, resulting in identification of 150 Ad-proteins as true positives. Similar analyses on SPOT-BioID results (data from SPOT-BioID for cytosolic proteins from the previous report (ref. 7) was used as a control to the one we used in this study) identified 587 proteins as Bt-proteins to be true positives. We then identified 13 proteins appearing in both ratiometrically analyzed results as Ad and Bt dual-labeled ones, resulting in the exclusion of six proteins from the initial 21 proteins (see the Venn diagram below and the list of proteins in an Excel file for Reviewer Only). The ratiometric analysis seemed to have removed false positives. However, upon careful analysis of the results, we noted the exclusion of five well-known MAM proteins—CLCC1, LBR, EMD, DNMI1L, and VDAC1—along with our newly identified protein, KPYM whose localization was confirmed by 3D rendering imaging and organelle fractionation, **Figures S8 and S9**). Their omission led to false negatives in the final result (See Figure below and attached Excel file for reviewer only).

Although ratiometric analysis proves beneficial in eliminating false positives among enriched proteins identified through conventional mass techniques, our results showed that it poses the risk of removing target proteins, resulting in false negatives that are challenging to confirm, especially given the limited availability of

alternative proteomics tools, particularly for spatiotemporal proteins.

It is important to highlight that the protein list in the original manuscript resulted from the intersection of proteomic data obtained by concurrently applying two different proteomic methods (SPOT-BioID and SPOT-SupraID) in the same experimental batch. This combination of methods, orthogonal in both protein enrichment and analysis, allows us to focus on identifying proteins highly localized to a specific cellular region, effectively minimizing false positives.

As no single method addresses all these challenges, we believe that our new method serves as a valuable addition to spatial-proteomics, providing complementary proteomic information useful for comprehending the intricate spatiotemporal connections of proteins during organelle communications. Therefore, we have decided to exclude the ratiometric analysis in the manuscript. Instead, we have now included a more detailed explanation on the analysis of our data in the revised manuscript to avoid confusion. I hope the reviewer will take this into account.

"[Redacted]"

2. Only ER transmembrane proteins could be detected by this method, introducing false negatives in the ER-mito contact site proteome. For instance, lipid-anchored proteins or cytosolic proteins enriched at the ER-mito contact site would not be captured and identified.

→ We appreciate the reviewer's insightful comments on the limitations of our method. As the reviewer pointed out, OrthoID may not capture the full spectrum of the ER-mito contact site proteome. Our method indeed focuses on ER transmembrane proteins, potentially missing lipid-anchored or cytosolic proteins localized at this interface. Given these limitations, the main point that we would like to emphasize in our study is to showcase OrthoID as a robust tool specifically tailored for the accurate identification of MAM proteins, notably those residing on the ER membrane (ERM) in this investigation.

A key strength of OrthoID is the simultaneous use of two enzymes without interference to each other in protein enrichment and analysis. It provides a novel research platform with modularity and flexibility in enzyme expression across various locations, orientations, and organelles on demand, enabled by typical genetic engineering. This adaptability positions OrthoID as a platform with the potential to address the reviewer's concern in principle, regarding protein capture at specific subcellular locations, including ER-PM, nucleus-vacuole, and mitochondria-lipid droplets.

Leveraging this potential for improvement and aiming to provide a balanced view of our methodology, we have now revised the conclusion of the manuscript to better reflect these considerations as follows: “*Notably, our experimental design strategically focuses on proteins at a specific spatial region, such as ERM at the ER-mito junction, which probably cannot cover the full spectrum of proteins at ER-mito junctions. In principle, enzyme expression can be modulated in various locations, orientations, and organelles on demand, offering modularity and flexibility of OrthoID. This feature underscores its great potential in pinpointing proteins at different contact sites, including ER-PM, nucleus-vacuole, and mitochondria-lipid droplets, significantly advancing our understanding of organelle interactions.*”

3. There are previous publications describing the proteome of the ER-mito contact site using different approaches. The authors should systematically and quantitatively compare the proteomic results. Are previously described ER-mito contact proteins detected in the current study? If there are discrepancies, the authors should thoroughly discuss potential reasons.

→ We appreciate the reviewer's suggestion to systematically and quantitatively compare our proteomic results with those from other methods focusing on ER-mito contact site proteomes. It's important to note that our method, OrthoID, emphasizes a highly focused investigation, targeting ERM proteins spatially localized at ER-mito contacts. This methodological distinction, utilizing a combination of two orthogonal proteomic approaches which significantly restricts the experimental conditions to focus on the identification of ERM proteins localized at the ER-mito contact, contrasts with other methods that predominantly detect cytosolic proteins around the ER-mito contact site, making a direct quantitative comparison between ours and others challenging.

Nevertheless, to do our best to address the reviewer's concern, we conducted a comparison analysis, at least, in a qualitative manner, after thoroughly reviewing the previous literature regarding MAM protein identification. The compiled data (Excel file S2) presents a clear overview of overlaps and discrepancies between MAM proteins detected by our system and others. Considering the methodological differences, we proceeded to discuss the divergences in our findings compared to those of other studies. For instance, while the overlapping list of MAM proteins between our findings and previous reports was mainly annotated as ERM or ERM-OMM dual-localization, novel proteins exclusively identified in OrthoID included ER luminal proteins such as KP YM, MESD, and GANAB. This distinction might be linked to the localized orientations of PL enzymes in OrthoID: APEX2 on ERM directing toward the ER lumen, and TurboID on OMM oriented toward the cytosol. This experimental setup differs from typical PL-based systems, enabling OrthoID to target ER proteins more selectively than other methods.

Furthermore, despite not detecting certain proteins found in other studies, such as IP₃R, OrthoID successfully identified their interacting partners like GANAB and KP YM. This suggests that OrthoID proficiently identifies MAM proteins associated with the ER-mito junctional complex focusing on ER proteins at ER-mito contact sites.

Considering this limitation and the methodological differences, we have now included discussions on the differences in methodological design and the resulting discrepancies between the proteins identified by our method and those by others, as follows. “*Although the proteins which are already reported as MAM proteins in the previous studies are predominantly annotated as ERM or ERM-OMM dual-localization (Excel file S2), the ones detected as MAM proteins for the first time by OrthoID such as KP YM, MESD, and GANAB, are ER luminal proteins. This distinction might be linked to the localizations and orientations of the PL enzymes in OrthoID:*

APEX2 expressed on ERM points towards the ER lumen, while TurboID on OMM is oriented towards cytosol. This stands in contrast to other PL-based systems that typically use only one PL enzyme expressed towards the cytosol, allowing OrthoID to target ER proteins more selectively.” and “Although OrthoID did not detect certain proteins identified as MAM proteins in other studies, such as IP₃R, it identifies their interacting partners such as GANAB and KP YM. This suggests that OrthoID is proficient in identifying MAM proteins associated with the ER-mito junctional complex. Therefore, the combination of a mutually orthogonal dual proteomics approach and binding pair systems, such as SPOT-SupraID and SPOT-BioID, has enabled the identification of proteins located in a specific spatial region, particularly at the ER-mito junction. In this study, OrthoID mainly focuses on an ER-oriented view. Nevertheless, its ability to detect not only the known MAM proteins, but also additional ER luminal proteins that transit the MAM can contribute to advancing our understanding of protein dynamics at the ER-mitochondria junction and the complex proteomic landscape at the vital cellular interface.”

4. Like any other proximity labeling or general pull-down experiments, **negative controls** are crucial for interpreting the specificity of the method and results. The proteomic design and mass spectrometry results are not fully clear here. The authors should describe whether **non-labeling and single-labeling** (biotinylation only and adamantylation only) controls were done and how they were used in data analysis and interpretation.

→ We appreciate the reviewer’s critical comments. Unlike conventional workflow, which requires negative controls to eliminate the proteins identified by non-specific interaction with target proteins, our method utilizing SPOT-ID method is fundamentally different from conventional PL-based proteomics as it selectively detects intentionally labeled peptides with Ad and Bt. This peptide-level analysis inherently ensures that only labeled proteins are included in our final protein list, effectively minimizing the potential for false positives. Also, we further enhanced the specificity of our results by data intersection from the parallel experiment of SPOT-BioID and SPOT-SupraID, identifying proteins labeled with both Ad and Bt (refer to our response to the first comment of this reviewer). Then, this dual labeling is subsequently validated through protein-level tandem enrichment, as illustrated in **Figure S7** (see below) in the revised supporting information.

To avoid confusion, we have incorporated a detailed scheme of the OrthoID workflow in **Figure S1** and additional explanation in the revised manuscript as follows. “*Notably, tandem mass spectrometry, which detects chemically labeled peptides with high reliability, ensures the selective identification of proteins influenced by the orientation and localization of PL-enzymes in proteomic analysis. This feature significantly diminishes false positives, thereby enhancing the accuracy of spatial proteomics.*” This description will assist in understanding the unique aspects and advantages of our approach for a simple experimental process while providing a focused and accurate identification of proteins relevant to ER-mito communication.

Figure S1. Schematic description for detailed process of OrthoID.

5. The functional studies are preliminary. The study of LRC59 is not fully convincing, particularly for the confocal imaging part. The authors should use electron microscopy and/or super-resolution microscopy to better examine whether LRC59 influences ER-mito contact formation or not.

→ In response to the reviewer's suggestion, we performed TEM analysis, which confirmed a decreased distance between the ER and mito, along with reduced ER-mito contact length in LRC59 knockdown cells (see below, **Figure 3C-E**). This TEM analysis validates our initial findings obtained through confocal imaging (**Figure 3A and 3B**) and emphasizes the involvement of LRC59 in the formation of ER-mito contact. Furthermore, we confirmed this finding through cross-validation by conducting an additional experiment using a split green fluorescent protein-based contact site sensor (SPLICS), demonstrating consistent knockdown-dependent changes in GFP signals, associated with LRC59 (see below, **Figure S12**). These additional result from TEM and GFP-based assessments strengthen our findings and have now been incorporated into the revised manuscript as follows. "By transmission electron microscopy (TEM), we detected a reduced distance between the ER and mitochondria, along with diminished ER-mito contact length in LRC59 knockdown cells (**Figure 3 c-e**). Additionally, we employed a split green fluorescent protein-based contact site sensor (SPLICS), utilizing one GFP half expressed

on ERM and the other on OMM. By CLSM, GFP signals were clearly observed from the control cells as a result of forming a complex of GFP fragments by ER-mito contact. However, these signals notably decreased in LRC59 knockdown cells, followed by signal recovery upon re-expression of LRC59 (**Figure S12**). These results strongly suggest the involvement of LRC59 in the formation and maintenance of ER-mito contact sites.”

Figure 3. Biological validations of LRC59 protein by ER-mito colocalization assay, TEM analysis, and calcium signaling assay.

(A) Colocalization assay between the ER and mitochondria with normal, LRC59 knockdown, and re-expressed cells by CLSM with TOM20-mRFP (red) as an OMM marker and Sec61B-mEmerald (green) as an ERM marker. Scale bar: 10 μm.

(B) Quantitative analysis of colocalization between ER and mitochondria using Manders' coefficient. (n = 110–135 cells, *** P < 0.001, **** P < 0.0001, the error bar represents the mean ± SD).

(C) TEM images of wild type (WT) and LRC59 knockdown (LRC59 KD) cells (yellow: mitochondria, red: ER). (D–E) Statistical analysis of TEM images. (D) Length of ER-mito contact site (E) Distance between ER and mitochondria (the analyses were performed by mapping ER and mitochondria from 10–11 cells for each group, *** P < 0.001, **** P < 0.0001, the error bar represents the mean ± SD)

(F–G) Changes in fluorescence intensity of calcium indicator protein. (F) RCEPIA1-er and (G) GCaMP6mt after histamine treatment in normal, LRC59 knockdown, and re-expressed cells normalized to the basal signals (F₀).

(H–I) Bar graphs (mean \pm SD) representing the peak amplitude of $\Delta F/F_0$ from (H) RCEPIA1-er and (I) GCaMP6mt in normal, LRC59 knockdown, and re-expressed cells (n.s. not significant, *** $P < 0.001$, **** $P < 0.0001$, the error bar represents the mean \pm SD, number represents the number of cells used for statistical analysis).

Figure S12. Effect of LRC59 in the formation of ER-mito contact site

(A) CLSM images of a split green fluorescent protein-based contact site sensor (SPLICS) expressed cells in normal, LRC59 knockdown, and LRC59 re-expressed conditions. ER-Short β_{11} was expressed in the ERM and OMM-GFP₁₋₁₀ was expressed in the OMM. BFP was expressed as a cell marker. Scale bar: 10 μ m.

(B) Quantitative analysis of the fluorescence intensity from SPLICSs (n = 250–280 cells, **** $P < 0.0001$, the error bar represents the mean \pm SD)

6. The mitophagy data is largely limited to proteomic profiling. Expression validation and/or functional investigation are highly encouraged. The authors made a few claims on ATD3B and MIC60, which requires further evidence.

→ We appreciate the insightful feedback from the reviewer. In our original manuscript, we presented evidence suggesting the involvement of IMM proteins, particularly ATD3B and MIC60, in mitophagy. These proteins were exclusively identified in CCCP-treated cells through our proteomic profiling. Considering the radius-dependent labeling feature of APEX2, known to be more effective for closer proteins, we hypothesized that the reduced ERM-OMM-IMM proximity during CCCP-induced mitophagy could facilitate the labeling of IMM proteins with Ad radicals, resulting in the identification of IMM proteins in our proteomic analysis. To validate this hypothesis, we performed a proximity ligation assay (PLA), allowing sensitive *in situ* detection of protein interactions through the appearance of fluorescent dots. The result from the assay confirmed a closer membrane proximity during CCCP-induced mitophagy, observed by a significantly larger number of the dots in the CCCP-treated cells (see below). This outcome provides additional support for our findings. We have now included this supplementary data

in **Figure S16** along with experimental details in the revised supplementary information and revised manuscript as follows. “Subsequently, we investigated the proximity between ERM-OMM and OMM-IMM by examining the interactions among specific membrane proteins. Using *IP₃R* and *VDAC* as markers for ERM and OMM interactions, respectively, and *SAM50* and *MIC60* (identified in this study) for OMM and IMM interactions, respectively, we conducted a proximity ligation assay (PLA) both with and without CCCP treatment. In CCCP-treated cells, the PLA results revealed an increased count of interactions among these membrane proteins, evident from a greater number of fluorescent PLA dots in CCCP-treated cells than in non-treated (**Figure S16**). This increase implies a closer membrane proximity, facilitating the labeling of IMM proteins with Ad radicals. These findings not only confirm our earlier proteomic profiling results, but also may provide insights into spatiotemporal cellular events occurring at membrane interfaces.”

Figure S16. Increased interactions between proteins at ERM, OMM and IMM detected by proximity ligation assay (PLA) under CLSM.

(A) Representative CLSM images and statistical analysis for counting the number of PLA dots in non-treated and CCCP-treated cells. Reaction performed between *IP₃R* as an ERM marker protein and *VDAC* as an OMM marker protein. (n = 180–190 cells, **** P < 0.0001, the error bar represents the mean ± SD, Scale bar: 10 μm)

(B) Representative CLSM images and statistical analysis for counting the number of PLA dots in non-treated and CCCP-treated cells. Reaction performed between *MIC60* as an IMM marker protein and *SAM50* as an OMM marker protein. (n = 80–100 cells, **** P < 0.0001, the error bar represents the mean ± SD, Scale bar: 10 μm)

REVIEWERS' COMMENTS

Reviewer #2 (Remarks to the Author):

The authors have carefully considered my concerns and have addressed them in full.

Reviewer #3 (Remarks to the Author):

The authors have fully addressed concerns. The paper is recommended for immediate publication.